# Responsive Noise-Relaying Diffusion Policy: Responsive and Efficient Visuomotor Control

**Zhuoqun Chen**\*                                                    *zhc057@ucsd.edu*
*UC San Diego*

**Xiu Yuan**\*                                                    *x1yuan@ucsd.edu*
*UC San Diego*

**Tongzhou Mu**                                                    *t3mu@ucsd.edu*
*UC San Diego*

**Hao Su**                                                    *haosu@ucsd.edu*
*UC San Diego*

**Reviewed on OpenReview:** *https://openreview.net/forum?id=LLWJkR6gaI*

## Abstract

Imitation learning is an efficient method for teaching robots a variety of tasks. Diffusion Policy, which uses a conditional denoising diffusion process to generate actions, has demonstrated superior performance, particularly in learning from multi-modal demonstrates. However, it relies on executing multiple actions predicted from the same inference step to retain performance and prevent mode bouncing, which limits its responsiveness, as actions are not conditioned on the most recent observations. To address this, we introduce Responsive Noise-Relaying Diffusion Policy (RNR-DP), which maintains a noise-relaying buffer with progressively increasing noise levels and employs a sequential denoising mechanism that generates immediate, noise-free actions at the head of the sequence, while appending noisy actions at the tail. This ensures that actions are responsive and conditioned on the latest observations, while maintaining motion consistency through the noise-relaying buffer. This design enables the handling of tasks requiring responsive control, and accelerates action generation by reusing denoising steps. Experiments on response-sensitive tasks demonstrate that, compared to Diffusion Policy, ours achieves 18% improvement in success rate. Further evaluation on regular tasks demonstrates that RNR-DP also exceeds the best acceleration method (DDIM) by 6.9% in success rate, highlighting its computational efficiency advantage in scenarios where responsiveness is less critical. Our project page is available at https://rnr-dp.github.io/.

## 1 Introduction

Imitation learning is a powerful approach for training robots to perform complex tasks, including grasping (Johns, 2021; Xie et al., 2020; Stepputtis et al., 2020), legged locomotion (Ratliff et al., 2007; Al-Hafez et al., 2023; Yang et al., 2023b), dexterous manipulation (Qin et al., 2022; Radosavovic et al., 2021), and mobile manipulation (Wong et al., 2022; Du et al., 2022). Advances in computer vision and natural language processing have led to increasingly sophisticated imitation learning frameworks, achieving impressive success across diverse tasks (Chen et al., 2021; Abramson, 1970; Florence et al., 2022; Shafiullah et al., 2022). A notable breakthrough, Diffusion Policy (Chi et al., 2023), models robot action sequences through a conditional denoising diffusion process, setting new benchmarks over traditional imitation learning techniques.

---

\*These authors contributed equally to this work.

Consequently, Diffusion Policy has rapidly gained traction and is now widely adopted as a foundational framework in both research and real-world applications.

However, Diffusion Policy faces significant limitations. As shown in Chi et al. (2023), its performance heavily depends on having a relatively large action horizon, $T_a$, which corresponds to executing multiple actions in the environment. The policy widely achieves optimal performance at $T_a = 8$ but suffers substantial degradation when $T_a = 1$, where only a single action is executed per inference. We attribute this to the nature of modeling multi-modal data: each inference independently samples actions aligned with a specific mode. Consequently, a larger action horizon is essential to ensure a sequence of actions adheres to the same mode, maintaining consistency. Conversely, using $T_a = 1$ leads to severe mode bouncing, causing significant performance drops. However, employing a large action horizon (e.g., $T_a = 8$) also introduces drawbacks, as most actions are not conditioned on the latest observations, thereby reducing responsiveness and adaptability to environmental changes. Empirically, we observe that Diffusion Policy struggles particularly with tasks requiring responsive control, such as handling dynamic objects.

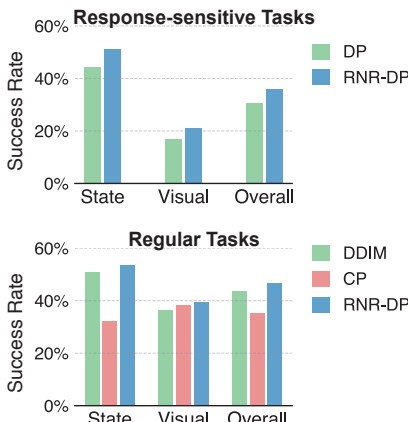

Figure 1: Our RNR-DP consistently delivers responsive and efficient control.

To address these issues, we propose Responsive Noise-Relaying Diffusion Policy (RNR-DP). We define *responsiveness* as the degree to which the current action leverages the most recent observation. Our approach fundamentally relies on a noise-relaying buffer that contains increasing noise levels and implements a sequential denoising mechanism. After each denoising step, conditioned on the most recent observations, the model executes an immediate noise-free action at the buffer's head and appends fully noisy actions at the buffer's tail. The noise-relaying buffer, combined with the sequential denoising mechanism, not only reuses denoising steps from previous outputs—allowing one denoising step to generate one action—but also ensures active control based on the most recent observations and prevents frequent mode bouncing, maintaining action consistency throughout the entire process. We name it **Responsive Noise-Relaying Diffusion Policy** because it employs a **noise-relaying buffer** at its core and has the ability to **respond quickly and actively to the environment**.

We evaluate our approach across a range of benchmarks, focusing on 9 tasks from three well-established datasets: ManiSkill2 (Gu et al., 2023), ManiSkill3 (Tao et al., 2024), and Adroit (Rajeswaran et al., 2017). Our primary evaluation targets 5 tasks involving dynamic object manipulation that demand responsive control. Empirical results show that RNR-DP significantly outperforms Diffusion Policy, delivering much more responsive control. Additionally, we extend our evaluation to tasks that do not require responsive control. The results indicate that, even on these simpler tasks, RNR-DP functions as a superior acceleration method compared to popular alternatives such as DDIM (Song et al., 2020) and Consistency Policy (Song et al., 2023; Prasad et al., 2024). Overall, our evaluations systematically demonstrate that RNR-DP provides both highly responsive and efficient control.

To summarize, our contributions are as follows:

- We identify a key limitation of Diffusion Policy: its reliance on a relatively large action horizon $T_a$, which compromises its responsiveness and adaptability to environmental changes.

- We propose Responsive Noise-Relaying Diffusion Policy (RNR-DP) which maintains a noise-relaying buffer with progressively increasing noise levels and employs a sequential denoising mechanism.

- We conduct extensive experiments on 5 tasks involving dynamic object manipulation and 4 simpler tasks that do not demand responsive control. The results consistently demonstrate that RNR-DP delivers both highly responsive and efficient control.

## 2 Related Work

**Diffusion Model Acceleration Techniques** Diffusion models (Ho et al., 2020) have garnered significant attention for their capacity to model complex distributions. However, their iterative sampling processes can be computationally expensive due to the large number of diffusion steps required. One approach to address the low inference speed of diffusion models is by reducing the number of denoising steps needed, as seen in works like (Song et al., 2020) and (Karras et al., 2022). Another line of research employs distillation-based techniques, which begin with a pretrained teacher model and train a new student model to take larger steps over the ODE trajectories that the teacher has already learned to map, as demonstrated in (Song et al., 2023), (Liu et al., 2023), and (Salimans & Ho, 2022). Among the most commonly used acceleration techniques in the robotics community are DDIM (Song et al., 2020) and Consistency Models (Song et al., 2023), which are particularly effective for speeding up the diffusion policy process.

**Diffusion Model for Motion Synthesis** Diffusion Model, renowned for its strong representational capabilities, has been widely applied to motion synthesis tasks (Shafir et al., 2023). Building on this foundation, Zhang et al. (2024) proposed an innovative framework that incorporates temporally varying denoising and maintains a motion buffer comprising progressively noised poses, enabling long-term motion synthesis. Inspired by TEDi, we develop a noise-relaying buffer with incrementally increasing noise levels and implement a sequential denoising mechanism to enhance Diffusion Policy, ensuring more efficient and responsive control.

**Diffusion Model as Policy** With the success of diffusion models in image synthesis and video generation (Ho et al., 2020), they have become a popular choice as policy backbones in the robotics community. These models are utilized in two main ways: 1) As policies in reinforcement learning (RL) methods, including offline RL (Wang et al., 2022; Hansen-Estruch et al., 2023; Mao et al., 2024), offline-to-online RL (Ding & Jin, 2023), and online RL (Yang et al., 2023a); 2) As policies in imitation learning (Chi et al., 2023; Reuss et al., 2023). Diffusion Policy belongs to the second category and has demonstrated state-of-the-art performance compared to other imitation learning methods (Shafiullah et al., 2022; Florence et al., 2022; Abramson, 1970). Furthermore, it exhibits significant potential for future research and practical applications (Yuan et al., 2024).

## 3 Background

**Diffusion Policy** (Chi et al., 2023) models control policies using Denoising Diffusion Probabilistic Models (DDPMs) (Ho et al., 2020), which have shown strong performance in generative modeling. In control, Diffusion Policy predicts the future action sequence $\mathbf{A}_t$ using a noise prediction network $\varepsilon_\theta(\mathbf{A}_t^{(k)}; \mathbf{O}_t, k)$, where $\mathbf{A}_t^{(k)} = \mathbf{A}_t + \epsilon^k$ is a perturbed version of the clean action sequence $\mathbf{A}_t$ with added Gaussian noise $\epsilon$ at noise level $k$. The model learns to estimate and remove noise by minimizing the mean squared error (MSE) loss:

$$\mathcal{L} = \|\varepsilon_\theta(\mathbf{A}_t^{(k)}; \mathbf{O}_t, k) - \varepsilon^k\|^2.$$

During inference, given an observation $\mathbf{O}_t$, the trained network iteratively refines the action sequence over $K$ denoising steps following:

$$\mathbf{A}_t^{(k-1)} = \alpha \left( \mathbf{A}_t^{(k)} - \gamma \varepsilon_\theta(\mathbf{A}_t^{(k)}; \mathbf{O}_t, k) + \epsilon \right),$$

where the initial action sequence $\mathbf{A}_t$ is sampled from $\mathcal{N}(0, 1)$, and $\epsilon \sim \mathcal{N}(0, \sigma^2 I)$ represents Gaussian noise. The predefined noise schedule functions $\alpha, \gamma$, and $\sigma$ are part of the DDPM scheduler (Ho et al., 2020). Once denoised to $\mathbf{A}_t^{(0)}$, the agent executes the first $n = T_a$ future steps after time $t$.

Table 1: We compare the performance of Diffusion Policy under different action horizon $T_a$ with multi-modal data and RL data.

| Demo Type | Traj Num | Ta=1 | Ta=2 | Ta=4 | Ta=8 |
|---|---|---|---|---|---|
| Multi-Modal | 1000 | 0.78 | 0.93 | 0.95 | **0.96** |
| RL (Single-Modal) | 500 | **0.62** | 0.59 | 0.58 | 0.55 |

## 4 Limitations of Diffusion Policy

This section explores the key limitations of Diffusion Policy in detail. In summary, Diffusion Policy relies on a relatively large action horizon, $T_a$, to ensure a sequence of actions adheres to the same mode and avoid frequent mode bouncing. However, using a large action horizon results in most actions being unconditioned on the latest observations, thereby limiting the policy's responsiveness to environmental changes. Section 4.1 delves into why Diffusion Policy requires a large action horizon, while Section 4.2 examines how this impacts responsiveness.

### 4.1 Why Diffusion Policy Needs A Large Action Horizon?

As shown in Chi et al. (2023), Diffusion Policy performs best with a relatively large action horizon $T_a$ (e.g., $T_a = 8$), while $T_a = 1$ significantly degrades performance. This is because each action sequence is independently denoised from noise, and in multi-modal settings, $T_a = 1$ allows actions to switch modes, causing inconsistencies. Executing multiple actions within the same mode is crucial for stable performance. To validate this, we train Diffusion Policy on 500 single-modal demonstrations from an RL agent in the ManiSkill2 StackCube task. Results in Table 1 show no performance drop with $T_a = 1$, even slightly improving over $T_a = 8$, confirming that mode bouncing in multi-modal settings is the main reason for requiring a large action horizon.

### 4.2 How A Large Action Horizon Limits Responsiveness?

As discussed above, a large action horizon $T_a$ is crucial for Diffusion Policy, particularly when modeling multi-modal data. However, an excessively large $T_a$ can hinder responsiveness to rapid environmental changes. Consider a dexterous robotic hand tasked with picking up a ball from a surface and transporting it to a goal position. This task demands continuous fine-grained control and rapid adaptation to unforeseen disturbances, such as the ball slipping or shifting unpredictably within the fingers. If the policy commits to executing $T_a = 8$ future actions, it may struggle to react promptly to subtle variations in grip force or contact dynamics. For example, if the ball begins to slip, a long action sequence could delay corrective actions, making recovery difficult. In contrast, a short action horizon ($T_a = 1$) allows the policy to continuously refine its grip based on real-time feedback, ensuring stable and controlled relocation. This example highlights that in contact-rich object manipulation tasks, a large action horizon forces the policy to predict complex interactions prematurely, making precise control more challenging. Empirically, we observe that Diffusion Policy struggles with such dynamic, contact-rich manipulation tasks, where real-time adaptability is essential, further underscoring the dilemma between long-horizon consistency and responsiveness.

## 5 Responsive Noise-Relaying Diffusion Policy

To achieve responsive and efficient control, our proposed approach relies on a noise-relaying buffer and the implemented sequential denoising machanism at its core (Section 5.1) with additional key design choices (Section 5.2). More details of the implementation including policy architecture and hyperparameters are summarized in Appendix B.

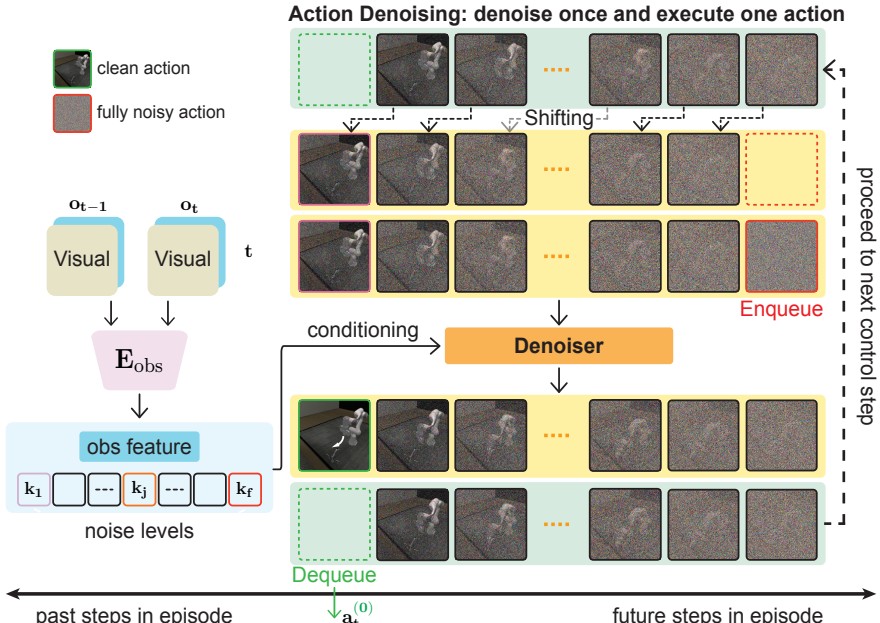

Figure 2: **Inference Overview of Responsive Noise-Relaying Diffusion Policy (RNR-DP).** The core of RNR-DP is the noise-relaying buffer (Section 5.1) and it has 3 stages during the entire control-loop, as exemplified by the transition between time step $t$ and time step $t+1$, **(1)** The buffer contains noisy actions with increasing noise levels **(2)** After denoising once, each action in the buffer is denoised for one step, clean action at the buffer's head is removed and executed (**dequeue**) **(3)** The remaining noisy actions are left shifted for one slot and a fully noisy action is appended to the buffer's tail (**enqueue**). The conditioning data is discussed with more details in Section 5.2 and Appendix B.3.

## 5.1 Noise-Relaying Buffer

The noise-relaying buffer $\tilde{\mathbf{Q}}_t = \{\mathbf{a}_t^{(1)}, \mathbf{a}_{t+1}^{(2)}, \ldots, \mathbf{a}_{t+f-2}^{(f-1)}, \mathbf{a}_{t+f-1}^{(f)}\}$ contains a sequence of noisy actions with linearly increasing noise levels from 1 to $f$, where $f$ is the buffer capacity as well as the total number of noise levels. As shown in Figure 2, after each denoising step, the trained network transforms $\tilde{\mathbf{Q}}_t$ into $\mathbf{Q}_t = \{\mathbf{a}_t^{(0)}, \mathbf{a}_{t+1}^{(1)}, \ldots, \mathbf{a}_{t+f-2}^{(f-2)}, \mathbf{a}_{t+f-1}^{(f-1)}\}$, producing a noise-free action $\mathbf{a}_t^{(0)}$ at the head. This clean action is immediately executed, and a fully noisy action is appended to the buffer's tail. For the next step, the buffer reuses $f-1$ denoising steps from previous outputs, ensuring consistency and avoiding full denoising from scratch. This sequential denoising mechanism conditions clean actions on the latest observations, enabling responsive and long-term active control. Pseudocode is provided in Algorithm 1.

## 5.2 Key Design Choices

**Mixture Noise Scheduling** We train the denoiser network following the DDPM (Ho et al., 2020) framework, and allow each action in $\mathbf{A}_t$ perturbed by independent noise levels. Given a fixed variance schedule $\beta_1, \ldots, \beta_f$ ($\beta_1 < \beta_f$), any $\mathbf{a}_j$ in $\mathbf{A}_t$ can be perturbed by one of the $f$ levels. During training, we use a mixed per-action noise injection scheme (*mixture schedule*): with probability $p_{\text{linear}}$, actions are perturbed by linearly increasing variances (*linear schedule*); with probability $1 - p_{\text{linear}}$, actions are perturbed by random variances from $\beta_1$ to $\beta_f$ (*random schedule*). The *random schedule* trains the model to denoise actions independently $p_\theta(\mathbf{a}^{(k-1)} \mid \mathbf{a}^{(k)}; \mathbf{O}), 1 \le k \le f$, while the *linear schedule* ensures smooth transitions across consecutive actions during inference. Unlike Diffusion Policy, which applies a single variance level to all actions in $\mathbf{A}_t$ per iteration, our mixture schedule enables diverse and robust training. An illustrative visualization is included in Appendix I.

**Laddering Initialization** During inference, we use a noise-relaying buffer with $f$ noisy action frames, following Algorithm 1. Initially, the buffer contains $f$ fully noisy actions sampled from $\mathcal{N}(\mathbf{0}, \mathbf{I})$, i.e., $\tilde{\mathbf{Q}} =$

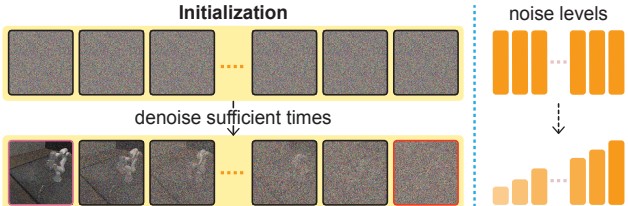

Figure 3: **A visulization of the initialization process.** Noise-relaying buffer contains only fully noisy actions sampled from standard multivariate Gaussian distribution of dimension $T_a$ at each action index before the initialization; the buffer is denoised for $f$ times until the buffer head contains minimal noise level and can be executed with one more model forward call.

$\{\mathbf{z}_j \sim \mathcal{N}(\mathbf{0}, \mathbf{I}) \mid j = 1, \ldots, f\}$. To align with training and avoid performance drops, we initialize the buffer by iteratively denoising it $f$ times using the *random schedule* conditioned on the initial observation $\mathbf{O}_0$. This transforms the buffer to follow the *linear schedule*, ensuring smooth and responsive control. Since this process transitions the buffer from uniform noise to monotonically increasing variances like a ladder, we term it *laddering initialization*, as illustrated in Figure 3. A more detailed illustrative visualization is included in Appendix J.

**Noise-Aware Conditioning** Unlike Diffusion Policy (Chi et al., 2023), which encodes a single diffusion step to one time embedding, our approach uses mixed scheduling and a noise-relaying buffer to handle multiple noise levels, requiring awareness of multiple diffusion steps. For each action's noise level $k_j$ ($1 \leq k_j \leq f$), we use an MLP to encode a time embedding, yielding $f$ embeddings. These are appended to the observation features from the encoder $E_{\mathrm{obs}}$. This allows the buffer, during both training and inference, to decode actions at each noise level using up-to-date observations, enabling more dynamic and consistent behaviors. See Appendix B.3 for more details.

## 6 Experiments

The goal of our experimental evaluation is to study the following questions:

1. Can Responsive Noise-Relaying Diffusion Policy outperform Diffusion Policy by delivering more responsive control? (Section 6.3)?

2. Can Responsive Noise-Relaying Diffusion Policy function as a superior acceleration method compared to commonly used alternatives on simpler tasks that do not require responsive control (Section 6.4)?

3. What are the effects of the components introduced by Responsive Noise-Relaying Diffusion Policy (Section 6.5)?

### 6.1 Experimental Setup

To validate the responsiveness and efficiency of Responsive Noise-Relaying Diffusion Policy, our experimental setup incorporates *variations in the following dimensions*:

- **Task Types:** Stationary robot arm manipulation, mobile manipulation, dual-arm coordination, dexterous hand manipulation, articulated object manipulation, and high-precision tasks.

- **Demo Sources:** Teleoperation, Task and Motion Planning, RL, and Model Predictive Control.

- **Observation Modalities:** State observation (low-dim) and visual observation (high-dim).

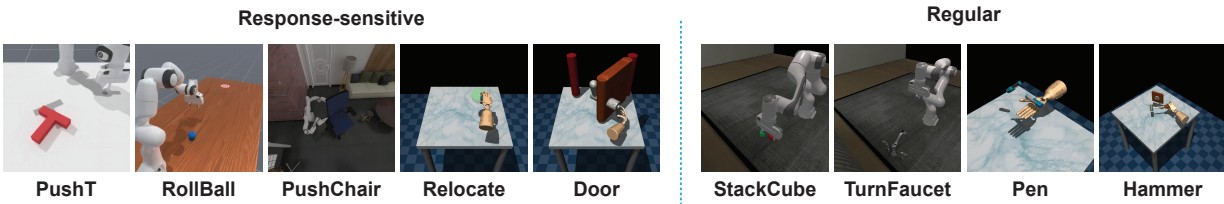

Figure 4: **Task Visualization in Simulation.** ManiSkill tasks including PushT, RollBall, PushChair, StackCube, TurnFaucet; Adroit tasks including Relocate, Door, Pen, Hammer. See Appendix A.1 for more properties about each task.

### 6.1.1 Task Descriptions

Our experiments are conducted on 9 tasks from 3 benchmarks: ManiSkill2 (robotic manipulation; 3 tasks), ManiSkill3 (robotic manipulation; 2 tasks) and Adroit (dexterous manipulation; 4 tasks). These tasks are separated into 2 groups to validate the responsiveness and efficiency of Responsive Noise-Relaying Diffusion Policy. A detailed discussion of task groupings and classification criteria can be found in Appendix D.

**Response-sensitive Group: Tasks Involving Dynamic Object Manipulation Requiring Responsive Control** We consider 5 challenging tasks involving contact-rich, dynamic object manipulation to assess the responsiveness of RNR-DP. PushT requires using a stick to push a T-shaped block to a target location and orientation. RollBall involves pushing and rolling a ball to a randomized goal region. PushChair tests bimanual manipulation of articulated objects with variations. Adroit Relocate involves picking up a ball and moving it to a goal position, while Adroit Door requires unlocking and opening a door. See Figure 4 for task visualizations, and more details are included in Appendix A.1.

**Regular Group: Simpler Tasks that Do Not Require Responsive Control** We consider the rest 4 simpler tasks that do not require responsive control to validate the efficiency of RNR-DP. StackCube requires picking up a cube and stack it onto another cube. TurnFaucet uses a stationary arm to turn on faucets of various geometries and topology. Adroit Pen repositions the blue pen to match the orientation of the green target. Adroit Hammer picks up a hammer and drives a nail into a board. More details are included in Appendix A.1.

## 6.2 Baselines

We compare our Responsive Noise-Relaying Diffusion Policy against a set of strong baselines related to Diffusion Policy.

**Diffusion Policy (DDPM)** is the original setting from Chi et al. (2023), which utilizes a conditional Denoising Diffusion Probalistic Model with discrete time scheduling to generate actions. We use 100 DDPM denoising steps in our experiments.

**Diffusion Policy (EDM)** is introduced in Prasad et al. (2024) to train a teacher model and employs the EDM (Karras et al., 2022) framework with continuous-time scheduling. This model takes the current position $x_t$, time $t$, and conditioning $o$ as inputs along a PFODE and is used to estimate the derivative of the PFODE's trajectory. We utilize the EDM model with Heun's second-order solver taking 80 denoising steps, which requires two neural network evaluations per discretized step in the ODE, resulting in a total of 159 neural function evaluations (**NFEs**).

**Diffusion Policy (DDIM)** is used in Chi et al. (2023) to accelerate Diffusion Policy with Diffusion Denoising Implicit Model framework (Song et al., 2020). We test 1, 2, 4, and 8 DDIM denoising steps.

**Consistency Policy (CP)** is introduced in Prasad et al. (2024), which employs the Consistency Trajectory Model (Kim et al., 2023) to distill the knowledge from a Teacher Model (EDM). We evaluate both the 1-step Consistency Policy and the 8-step chaining Consistency Policy.

Table 2: Evaluation on tasks requiring responsive control (Response-sensitive Group) from ManiSkill and Adroit benchmarks (State Observations). We report average success rate ($\uparrow$) of the best checkpoint for 1000 episodes across 10 random seeds. Results that are statistically better are highlighted in **bold**. Our results are highlighted in **light-blue** cells.

| Method | Relocate | Door | PushChair w/ g | PushChair w/o g | RollBall | PushT |
|---|---|---|---|---|---|---|
| DP | 0.422 | 0.558 | 0.495 | 0.635 | 0.083 | 0.470 |
| RNR-DP | **0.585** | **0.629** | **0.547** | **0.694** | **0.121** | **0.491** |

Table 3: Evaluation on tasks (Response-sensitive Group) from ManiSkill and Adroit benchmarks (Visual Observations). We report values under the same settings as in Table 2. Tasks in which none of the methods achieve a reasonable success rate under visual observations are omitted.

| Method | Door (Adroit) | RollBall (MS3) | PushT (MS3) |
|---|---|---|---|
| DP | 0.079 | 0.080 | 0.349 |
| RNR-DP | **0.122** | **0.131** | **0.379** |

**Streaming Diffusion Policy (SDP)** is a recent advancement over Diffusion Policy that stays close to our approach. See Appendix C.2 for detailed discussion.

### 6.3 Results & Analysis on Responsive Control

First, we provide comprehensive evaluation and validate the responsiveness of RNR-DP on 5 tasks (Response-sensitive Group) involving contact-rich dynamic object manipulation, a primary focus of our policy. Figure 1 shows an overview of performance improvement. As shown in Table 2 and Table 3, RNR-DP consistently outperforms Diffusion Policy across all tested scenarios in both state and visual experiments. Specifically, in state experiments, RNR-DP achieves a 15.1% improvement over Diffusion Policy, while in visual experiments, it demonstrates a 24.9% improvement. Overall, RNR-DP surpasses Diffusion Policy by 18.0%. Notably, in the Relocate task, RNR-DP achieves a significant performance boost of 38.6% over Diffusion Policy.

We attribute these significant performance gains to the design of our noise-relaying buffer and sequential denoising mechanism, which offer two key advantages. The noise-relaying buffer ensures that the denoising of each action remains consistent, allowing actions to follow the same mode. This effectively eliminates frequent mode bouncing, enabling RNR-DP to support single-action rollouts ($T_a = 1$). Furthermore, the single-action rollout in RNR-DP ensures that all actions are conditioned on the latest observations, resulting in significantly more responsive control to environmental changes compared to Diffusion Policy.

### 6.4 Results & Analysis on Efficient Control

In this section, we conduct extended experiments on 4 simpler tasks (Regular Group) that do not require responsive control to evaluate the efficiency of RNR-DP. An overview of the performance improvements is provided in Figure 1. Section 6.4.1 introduces a metric designed for fair efficiency comparisons, while Section 6.4.2 provides a detailed analysis of the empirical results compared to commonly used acceleration methods.

#### 6.4.1 Neural Function Evaluations per Action (NFEs/a)

Adopting the settings from Chi et al. (2023), Diffusion Policy and related methods utilize a relatively large action horizon ($T_a$) to achieve better performance, whereas RNR-DP employs a single-action rollout ($T_a = 1$) at each inference. To facilitate a fair comparison of efficiency, we introduce a new metric, **Neural Function Evaluations per Action (NFEs/a)**, as defined in Equation (1).

Table 4: Evaluation on simpler tasks (Regular Group) not requiring responsive control from ManiSkill and Adroit benchmarks (State Observations). We report average success rate (↑) and overall average success rate of all tasks (↑). Particularly, DDIMs with NFEs/a = 1, CPs with NFEs/a = 1 are highlighted in light-green, light-red cells respectively. Our results are highlighted in light-blue cells.

| Method | Steps (S) | NFEs/a | StackCube | TurnFaucet w/ g | TurnFaucet w/o g | Pen | Hammer | Avg. SR of tasks |
|---|---|---|---|---|---|---|---|---|
| DDPM | **100** | **12.5** | **0.960** | **0.495** | **0.595** | **0.508** | **0.120** | **0.536** |
| DDIM | 1 | 0.125 | 0.000 | 0.000 | 0.000 | 0.110 | 0.000 | 0.000 |
|  | 2 | 0.25 | 0.214 | 0.433 | 0.594 | 0.190 | 0.000 | 0.286 |
|  | 4 | 0.5 | 0.959 | 0.482 | 0.612 | 0.476 | 0.000 | 0.506 |
|  | 8 | 1 | **0.964** | 0.477 | **0.614** | 0.483 | 0.000 | 0.508 |
| EDM | 80 | 20 | 0.955 | 0.449 | 0.575 | 0.517 | 0.120 | 0.523 |
| CP | 1 | 0.125 | 0.214 | 0.051 | 0.028 | 0.326 | 0.000 | 0.124 |
|  | 8 | 1 | 0.680 | 0.112 | 0.299 | 0.483 | 0.041 | 0.323 |
| RNR-DP | 1 | 1 | 0.935 | **0.531** | 0.594 | **0.487** | **0.139** | **0.537** |

$$\text{NFEs/a} = \frac{\text{NFEs}}{T_a} \tag{1}$$

### 6.4.2 Empirical Comparison with Commonly Used Acceleration Methods

To evaluate the efficiency of Responsive Noise-Relaying Diffusion Policy (RNR-DP), we compare it against common acceleration methods, including DDIM (Chi et al., 2023) and Consistency Policy (Prasad et al., 2024), on simpler tasks that do not require responsive control (Regular Group). As shown in Table 4 and Appendix C.1, RNR-DP consistently achieves the highest overall performance among all DDIM and Consistency Policy variants. For a fair comparison, we specifically evaluate RNR-DP against 8-step DDIM (NFEs/a = 1) and 8-step-chaining Consistency Policy (NFEs/a = 1). In state-based experiments, RNR-DP outperforms 8-step DDIM by 5.7% and 8-step-chaining Consistency Policy by 66.2%. In vision-based experiments, it surpasses 8-step DDIM by 8.5% and 8-step-chaining Consistency Policy by 3.4%, yielding an overall improvement of 6.9% over 8-step DDIM and 32.0% over 8-step-chaining Consistency Policy. Additionally, both DDIM and Consistency Policy struggle in certain tasks. For instance, 8-step DDIM achieves a 0% success rate on Adroit Hammer, while Consistency Policy performs poorly on StackCube and TurnFaucet. In contrast, RNR-DP demonstrates consistent and robust performance across all tasks. Notably, RNR-DP achieves an average success rate comparable to Diffusion Policy across all state and visual experiments while being 12.5 times faster. These results confirm that even on simpler tasks, RNR-DP serves as a superior acceleration method, enabling efficient control.

We attribute this robust performance to the design of the noise-relaying buffer and sequential denoising mechanism. By reusing denoising steps from previous outputs, RNR-DP requires only one denoising step to generate a single action, while ensuring all actions undergo sufficient denoising to maintain high action quality. In contrast, DDIM and Consistency Policy significantly reduce the number of denoising steps per action, leading to pronounced performance drops.

### 6.5 Ablation Study

We conduct various ablations to provide further insights on the effects of components of RNR-DP.

**Noise Scheduling Scheme** We conduct a comprehensive study to evaluate how different noise scheduling schemes impact the performance of the Responsive Noise-Relaying Diffusion Policy. As shown in Table 5, the results demonstrate that relying solely on either a *linear schedule* or a *random schedule* significantly reduces task success rates. This highlights the importance of integrating *mixture scheduling* to fully exploit the potential of the noise-relaying buffer design. By combining the two schedules, our model effectively utilizes different noise levels, enhancing the robustness and adaptability of action generation.

Table 5: Ablation study on noise scheduling scheme during training (Section 5.2). Numbers represent average success rates (↑). Numbers in parenthesis indicate the performance drop after removing key component of our Responsive Noise-Relaying Diffusion Policy.

| Ablation | Relocate (Adroit) | Door (Adroit) |
|---|---|---|
| Linear | 0.323 (-26.2%) | 0.404 (-22.5%) |
| Random | 0.389 (-19.6%) | 0.360 (-26.9%) |
| Mixture (Ours) | **0.585** | **0.629** |

Table 6: Ablation study on noise-relaying buffer initialization (Section 5.2).

| Ablation | Relocate (Adroit) | Door (Adroit) |
|---|---|---|
| Pure noise | 0.522 (-6.3%) | 0.516 (-11.3%) |
| Laddering (Ours) | **0.585** | **0.629** |

Table 7: Ablation study on model prediction type (Section 5.2).

| Ablation | Relocate (Adroit) | Door (Adroit) |
|---|---|---|
| Action | 0.154 (-43.1%) | 0.374 (-25.5%) |
| Noise (Ours) | **0.585** | **0.629** |

**Noise-Relaying Buffer Initialization** We conduct an ablation study to evaluate the effectiveness of our initialization scheme. *Pure noise* directly uses a fully noisy buffer as the initialization for subsequent operations. As shown in Table 6, *laddering initialization* achieves significant performance gains compared to *pure noise*. This result highlights the critical role of our initialization scheme in enabling strong and robust performance.

**Noise-Relaying Buffer Capacity** We also evaluate the impact of the noise-relaying buffer capacity to the performance, the only tuned hyperparameter in our approach. As shown in Figure 5, on the Adroit Relocate task, RNR-DP achieves peak performance of 58.5% at a buffer capacity of 84 and maintains strong performance within the range of 56 to 92, demonstrating a wide tolerance. Performance declines when the buffer capacity is too small (e.g., 48) or too large (e.g., 100). Notably, Diffusion Policy achieves only 42.2%, while RNR-DP with buffer capacities between 48 and 100 consistently outperforms it.

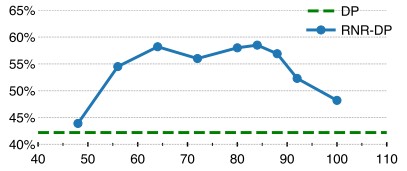

Figure 5: Noise-relaying buffer capacity v.s. average success rate.

This highlights that for a single task, the noise-relaying buffer offers robust performance over a broad range and is not difficult to tune. Empirically, starting with a value between 56 and 84 and making adjustments suffices most cases.

**Model Prediction Type** In addition, we investigate the impact of the model's prediction type. As shown in Table 7, models predicting the added noise component outperform those directly predicting the action sequence. This finding aligns with the common practice of using noise prediction in diffusion models within the vision domain (Ho et al., 2020).

## 7 Conclusion and Future Work

In this paper, we identify the key limitation of Diffusion Policy: its reliance on a relatively large action horizon $T_a$ compromises its responsiveness and adaptability to environment changes. To address these

issues, we propose Responsive Noise-Relaying Diffusion Policy which maintains a noise-relaying buffer with progressively increasing noise levels and employs a sequential denoising mechanism. Our method provides more responsive control than Diffusion Policy on 5 tasks involving dynamic object manipulation, and delivers more efficient control than commonly used acceleration methods on 4 simpler tasks.

**Limitations**. A limitation of this work is the lack of real-robot evalutions. We discuss our method's potential advantages in real-robot deployment in Appendix E. We will conduct complex real world deployment for future research.

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

# A   Further Details on the Experimental Setup

## A.1   Task Descriptions

We consider a total of 9 continuous control tasks from 3 benchmarks: ManiSkill2 (Gu et al., 2023), ManiSkill3 (Tao et al., 2024), and Adroit (Rajeswaran et al., 2017). This section provides detailed task descriptions on overall information, task difficulty, object sets, state space, and action space. Some task details are listed in Table 8.

Table 8: We consider 9 continuous tasks from 3 benchmarks. We list important task details below.

| Task | State Obs Dim $C_{\text{state}}$ | Act Dim $C_a$ | Max Episode Step |
|---|---|---|---|
| ManiSkill3: PushT | 31 | 7 | 100 |
| ManiSkill3: RollBall | 44 | 4 | 80 |
| ManiSkill2: StackCube | 55 | 4 | 200 |
| ManiSkill2: TurnFaucet | 43 | 7 | 200 |
| ManiSkill2: PushChair | 131 | 20 | 200 |
| Adroit: Door | 39 | 28 | 300 |
| Adroit: Pen | 46 | 24 | 200 |
| Adroit: Hammer | 46 | 26 | 400 |
| Adroit: Relocate | 39 | 30 | 400 |

### A.1.1   ManiSkill2 Tasks

**StackCube**

- Overall Description: Pick up a red cube and place it onto a green one. See Figure 6 for episode visualization.

- Task Difficulty: This task requires precise control. The gripper needs to firmly grasp the red cube and accurately place it onto the green one.

- Object Variations: No object variations

- Action Space: Delta position of the end-effector and joint positions of the gripper.

- State Observation Space: Proprioceptive robot state information, such as joint angles and velocities of the robot arm, and task-specific goal information

- Visual Observation Space: one 64x64 RGBD image from a base camera and one 64x64 RGBD image from a hand camera.

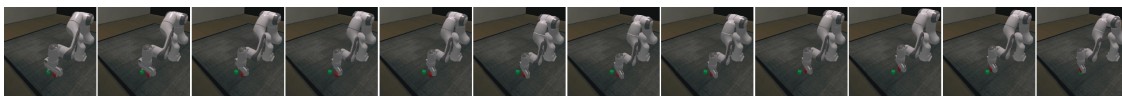

Figure 6: StackCube Episode Visualization.

**TurnFaucet**

- Overall Description: Turn on a faucet by rotating its handle.

- Task Difficulty: This task needs to handle object variations. See Figure 7 for episode visualization.

- Object Variations: We have a source environment containing 10 faucets, and the dataset is collected in the source environment. w/o g means the agent directly interacts with the source environment online; w/ g means the agent interacts with the target environment online, which contains 4 novel faucets.

- Action Space: Delta pose of the end-effector and joint positions of the gripper.

- State Observation Space: Proprioceptive robot state information, such as joint angles and velocities of the robot arm, the mobile base, and task-specific goal information.

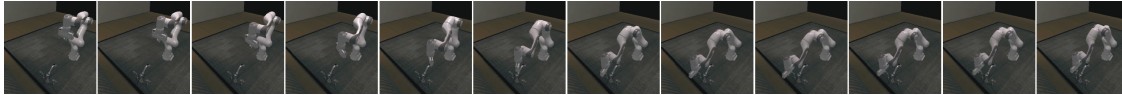

Figure 7: TurnFaucet Episode Visualization.

**PushChair**

- Overall Description: A dual-arm mobile robot needs to push a swivel chair to a target location on the ground (indicated by a red hemisphere) and prevent it from falling over. The friction and damping parameters for the chair joints are randomized. See Figure 8 for episode visualization.

- Task Difficulty: This task needs to handle object variations.

- Object Variations: We have a source environment containing 5 chairs, and the dataset is collected in the source environment. w/o g means the agent directly interacts with the source environment online; w/ g means the agent interacts with the target environment online, which contains 3 novel faucets.

- State Observation Space: Proprioceptive robot state information, such as joint angles and velocities of the robot arm, task-specific goal information.

- Visual Observation Space: three 50x125 RGBD images from three cameras 120° apart from each other mounted on the robot.

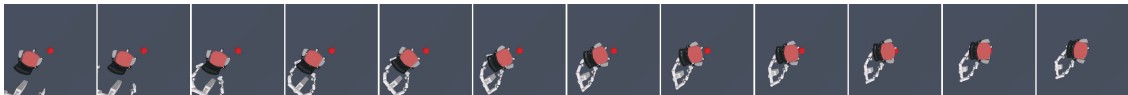

Figure 8: PushChair Episode Visualization.

### A.1.2 ManiSkill3 Tasks

**PushT**

- Overall Description: It is a simulated version of the real-world push-T task from Diffusion Policy: https://diffusion-policy.cs.columbia.edu/. In this task, the robot needs to precisely push the T-shaped block into the target region, and move the end-effector to the end-zone which terminates the episodes. The success condition is that the T block covers 90% of the 2D goal T's area. See Figure 9 for episode visualization.

- Task Difficulty: The task involves manipulating a dynamic T-shaped object, which introduces non-linear dynamics, friction, and contact forces.

- Object Variations: No object variations.

- Action Space: Delta pose of the end-effector and joint positions of the gripper.

- State Observation Space: Proprioceptive robot state information, such as joint angles and velocities of the robot arm, and task-specific goal information.

- Visual Observation Space: one 64x64 RGBD image from a base camera.

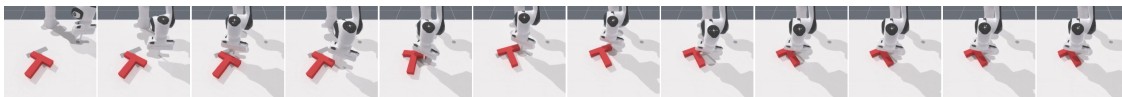

Figure 9: PushT Episode Visualization. The T-Block is pushed from sampled initial configuration to the goal area.

**RollBall**

- Overall Description: A task where the objective is to push and roll a ball to a goal region at the other end of the table. The success condition is that The ball's xy position is within goal radius (default 0.1) of the target's xy position by euclidean distance. See Figure 10 for episode visualization.

- Task Difficulty: The task involves manipulating a dynamic ball, which introduces non-linear dynamics, friction, and contact forces.

- Object Variations: No object variations.

- Action Space: Delta position of the end-effector and joint positions of the gripper.

- State Observation Space: Proprioceptive robot state information, such as joint angles and velocities of the robot arm, and task-specific goal information.

- Visual Observation Space: one 64x64 RGBD image from a base camera.

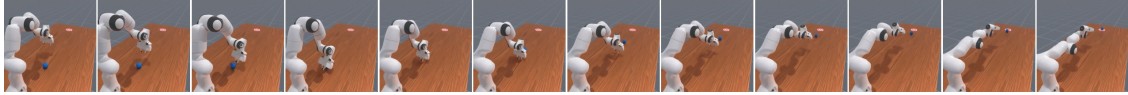

Figure 10: Rollball Episode Visualization. The blue ball is pushed and rolled from sampled initial configuration to the target red circle.

### A.1.3 Adroit Tasks

**Adroit Door**

- Overall Description: The environment is based on the Adroit manipulation platform, a 28 degree of freedom system which consists of a 24 degrees of freedom ShadowHand and a 4 degree of freedom arm. The task to be completed consists on undoing the latch and swing the door open. See Figure 11 for episode visualization.

- Task Difficulty: The latch has significant dry friction and a biass torque that forces the door to stay closed. No information about the latcch is explicitly provided. The position of the door is randomized.

- Object Variations: No object variations.

- Action Space: Absolute angular positions of the Adoit hand joints.

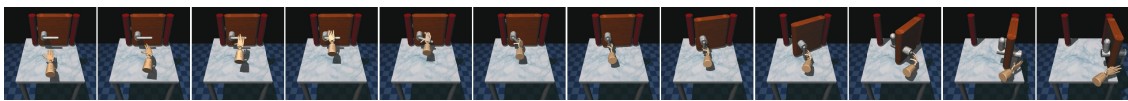

Figure 11: Door Episode Visualization.

- State Observation Space: The angular position of the finger joints, the pose of the palm of the hand, as well as state of the latch and door.

- Visual Observation Space: one 128x128 RGB image from a third-person view camera.

**Adroit Pen**

- Overall Description: The environment is based on the Adroit manipulation platform, a 28 degree of freedom system which consists of a 24 degrees of freedom ShadowHand and a 4 degree of freedom arm. The task to be completed consists on repositioning the blue pen to match the orientation of the green target. See Figure 12 for episode visualization.

- Task Difficulty: The target is also randomized to cover all configurations.

- Object Variations: No object variations.

- Action Space: Absolute angular positions of the Adroit hand joints.

- State Observation Space: The angular position of the finger joints, the pose of the palm of the hand, as well as the pose of the real pen and target goal.

- Visual Observation Space: one 128x128 RGB image from a third-person view camera.

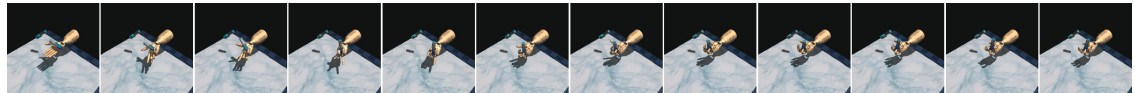

Figure 12: Pen Episode Visualization.

**Adroit Hammer**

- Overall Description: The environment is based on the Adroit manipulation platform, a 28 degree of freedom ShadowHand and a 4 degree of freedom arm. The task to be completed consists on picking up a hammer with and drive a nail into a board. See Figure 13 for episode visualization.

- Task Difficulty: The nail position is randomized and has dry friction capable of absorbing up to 15N force.

- Object Variations: No object variation.

- Action Space: Absolute angular positions of the Adroit hand joints.

- State Observation Space: The angular position of the finger joints, the pose of the palm of the hand, the pose of the hammer and nail, and external forces on the nail.

- Visual Observation Space: one 128x128 RGB image from a third-person view camera.

**Adroit Relocate**

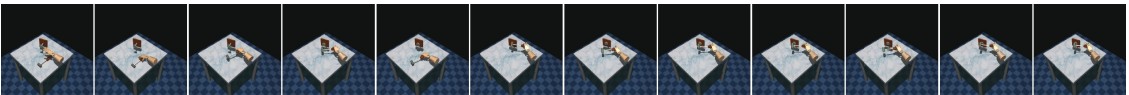

Figure 13: Hammer Episode Visualization.

- Overall Description: The environment is based on the Adroit manipulation platform, a 30 degree of freedom system which consists of a 24 degrees of freedom ShadowHand and a 6 degree of freedom arm. The task to be completed consists on moving the blue ball to the green target. See Figure 14 for episode visualization.

- Task Difficulty: The positions of the ball and target are randomized over the entire workspace.

- Object Variations: No object variations.

- Action Space: Absolute angular positions of the Adroit hand joints.

- State Observation Space: The angular position of the finger joints, the pose of the palm of the hand, as well as kinematic information about the ball and target.

- Visual Observation Space: one 128x128 RGB image from a third-person view camera.

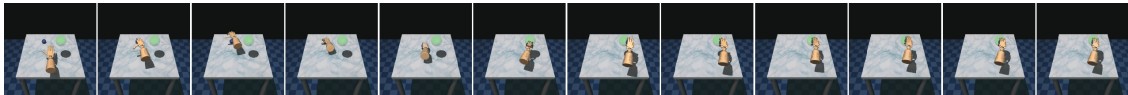

Figure 14: Relocate Episode Visualization.

## A.2 Demonstrations

This subsection provides the details of demonstrations used in our experiments. See Table 9. ManiSkill2 and ManiSkill3 demonstrations are provided in Gu et al. (2023) and Tao et al. (2024), and Adroit demonstrations are provided in Rajeswaran et al. (2017).

Table 9: Demonstration sources, numbers and generation methods.

| Task | Traj Num for Training | Generation Method |
|------|-----------------------|-------------------|
| ManiSkill3: PushT | 1000 | Reinforcement Learning |
| ManiSkill3: RollBall | 1000 | Reinforcement Learning |
| ManiSkill2: StackCube | 1000 | Task & Motion Planning |
| ManiSkill2: TurnFaucet | 1000 | Model Predictive Control |
| ManiSkill2: PushChair | 1000 | Reinforcement Learning |
| Adroit: Door | 25 | Human Demonstration |
| Adroit: Pen | 25 | Human Demonstration |
| Adroit: Hammer | 25 | Human Demonstration |
| Adroit: Relocate | 25 | Human Demonstration |

# B Implementation Details

## B.1 Noise-Relaying Diffusion Policy Inference

We summarize the inference pseudo-code of our RNR-DP in Algorithm 1.

---

**Algorithm 1** Noise-relaying Diffusion Policy Inference

---

1: **Require**: denoising model, $\varepsilon_\theta$; observation, $\mathbf{O}_t$; noise-relaying buffer, $\tilde{\mathbf{Q}}_t$; buffer capacity $f$;

2: **while** task execution **do**

3: $\quad \mathbf{Q}_t \leftarrow \varepsilon_\theta(\tilde{\mathbf{Q}}_t; \mathbf{O}_t, \{1, \cdots, f\}) \qquad\qquad\qquad\qquad \triangleright \varepsilon_\theta$ is trained using $f$ noise levels

4: $\quad \mathbf{a}_t^{(0)} \leftarrow \mathbf{Q}_t.\text{pop}(0) \qquad\qquad\qquad\qquad\qquad\qquad \triangleright \mathbf{a}_t^{(0)}$ is a clean action (fully denoised)

5: $\quad \tilde{\mathbf{Q}}_t \leftarrow \mathbf{Q}_t.\text{push}(\mathbf{z}) \qquad\qquad\qquad\qquad \triangleright \mathbf{z}$ is a random noisy action sampled from $\mathcal{N}(\mathbf{0}, \mathbf{I})$

6: $\quad \mathbf{O}_t \leftarrow \text{env.step}(\mathbf{a}_t^{(0)}) \qquad\qquad\qquad \triangleright$ executes $\mathbf{a}_t^{(0)}$ and envrionment updates observation

7: $\quad t \leftarrow t + 1 \qquad\qquad\qquad\qquad\qquad\qquad\qquad \triangleright$ update timestep for the next iteration

---

## B.2 Noise-Relaying Diffusion Policy Training

We summarize the training pseudo-code of our RNR-DP in Algorithm 2.

---

**Algorithm 2** Responsive Noise-Relaying Diffusion Policy Training

---

1: **Require**: demonstration dataset, $\mathcal{D} = \{(\mathbf{O}_i, \mathbf{A}_i)\}_{i=1}^N$; denoising model, $\varepsilon_\theta$; number of diffusion steps, $f$

2: **repeat**

3: $\quad$ Sample $(\mathbf{O}, \mathbf{A}) \sim \mathcal{D}$

4: $\quad$ Sample $p \sim \text{Unif}(0, 1)$; Sample noise $\boldsymbol{\epsilon} \sim \mathcal{N}(0, \mathbf{I})$ and reshape to $\mathbb{R}^{C_a \times f}$

5: $\quad$ **if** $p < p_{\text{linear}}$ :

6: $\quad\quad \mathbf{k} = \{k_1 = 1, \cdots, k_f = f\} \qquad\qquad\qquad\qquad\qquad\qquad \triangleright$ linear schedule

7: $\quad$ **else**

8: $\quad\quad \mathbf{k} = \{k_1 \sim \text{Unif}(\{1, \cdots, f\}), \cdots, k_f \sim \text{Unif}(\{1, \cdots, f\})\} \qquad \triangleright$ random schedule

9: $\quad$ **for all** $\mathbf{a}_j \in \mathbf{A}$ indexed by frame index $j$ **do**

10: $\quad\quad \hat{\mathbf{a}}_j = \sqrt{\bar{\alpha}_{t_j}} \mathbf{a}_j + \sqrt{1 - \bar{\alpha}_{t_j}} \boldsymbol{\epsilon}_j \qquad\qquad\qquad \triangleright$ perturbe each $\mathbf{a}_j$ independently

11: $\quad \hat{\mathbf{A}} = \{\hat{\mathbf{a}}_0, \cdots, \hat{\mathbf{a}}_{f-1}\}$

12: $\quad$ Take gradient descent step to update $\theta$ on

13: $\quad\quad \nabla_\theta \|\boldsymbol{\epsilon} - \varepsilon_\theta(\hat{\mathbf{A}}; \mathbf{O}, \mathbf{k})\| \qquad\qquad\qquad\qquad\qquad \triangleright$ noise-aware conditioning

14: **until** converged

---

## B.3 Policy Architecture

We build our RNR-DP on top of the UNet-based architecture of Diffusion Policy (Chi et al., 2023). The model includes 2 downsampling modules and 2 upsampling modules with each module containing 2 residual blocks. The residual block consists of 1D temporal convolutions (Conv1d), group normalizations (GN), and Mish activation layers. The encoded noise-aware conditioning data (Section 5.2) is fused into each residual block through the FiLM transformation (Perez et al., 2018). The raw conditioning data is of shape $R^{f \times (C_{\text{emb}} + C_{\text{state}})}$ for state policies and of shape $R^{f \times (C_{\text{emb}} + C_{\text{visual}} + C_{\text{state}})}$ for visual policies. See Figure 15 for the visualization of a visual policy. We follow the UNet denoiser design for the observation that transformer-based policies are more sensitive to hyperparameters and often require more tuning (Chi et al., 2023). The choice of policy architecture is orthogonal to our method and we believe our design would also improve this policy class.

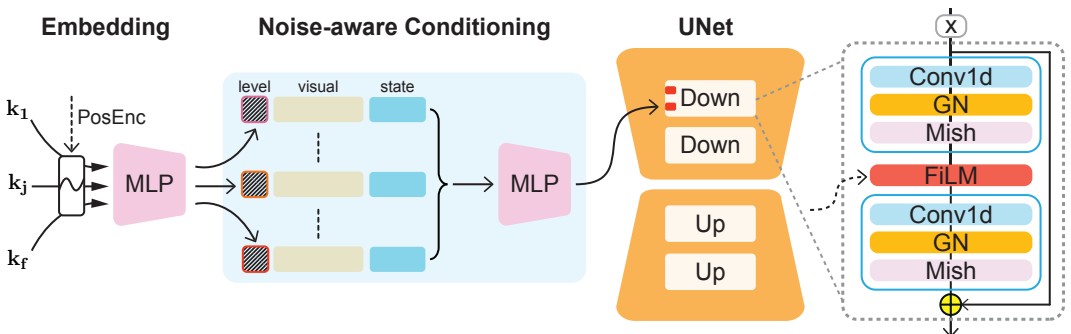

Figure 15: The detailed policy architecture for our RNR-DP. We only extract visual features for visual policies.

## B.4 Important Hyperparameters

### B.4.1 Key Hyperparameters of RNR-DP

We summarize the key hyperparameters of RNR-DP in Table 10. The observation horizon $T_o$ and noise-relaying buffer capacity $f$ for each task is listed in Table 11. The number of trainable parameters for each task is listed in Table 12.

Table 10: We list the key hyperparameters of RNR-DP used in our experiments.

| Hyperparameter | Value |
|---|---|
| RNR-DP Noise Scheduling Scheme | Mixture Sampling($p_{\text{linear}}$) $p_{\text{linear}} = 0.4$ |
| RNR-DP Model Prediction Type | Noise |
| Diffusion Step Embedding Dimension | 64 |
| UNet Downsampling Dimensions | $[64, 128, 256]$ |
| Optimizer | AdamW |
| Weight Decay | 1e-6 |
| Learning Rate | 1e-4 |
| Learning Rate Scheduler | Cosine |
| EMA Model Update | 0.9999 |
| Online Evaluation Episodes | 1000 |

### B.4.2 Key Hyperparameters of Diffusion Policy

We summarize the key hyperparameters of Diffusion Policy in Table 13. The observation horizon $T_o$, action execution horizon $T_a$ and action prediction horizon $T_p$ for each task are listed in Table 14.

## B.5 Training Details

We train our models and baselines with cluster assigned GPUs (NVIDIA 2080Ti & A10). We use AdamW optimizer with an initial learning rate of 1e-4, applying 500 warmup steps followed by cosine decay. We use batch size of 1024 for state policies and 256 for visual policies for both ManiSkill and Adroit benchmarks. We evaluate DP, CP and RNR-DP model checkpoints using EMA weights every 10K training iterations for ManiSkill tasks and every 5K for Adroit tasks. DDIMs are evaluated using the best checkpoints of DDPMs in an offline manner. CPs are trained using the best checkpoints of EDMs.

Table 11: The observation horizon and noise-relaying buffer capacity of our RNR-DP for each task.

| Task | Obs $T_o$ | Capacity $f$ |
|---|---|---|
| ManiSkill3: PushT (Visual) | 2 | 48 |
| ManiSkill3: RollBall (Visual) | 2 | 64 |
| ManiSkill2: StackCube (Visual) | 2 | 84 |
| Adroit: Pen (Visual) | 2 | 4 |
| Adroit: Hammer (Visual) | 2 | 64 |
| Adroit: Door (Visual) | 2 | 56 |
| | | |
| ManiSkill3: PushT (State) | 2 | 32 |
| ManiSkill3: RollBall (State) | 2 | 4 |
| ManiSkill2: StackCube (State) | 2 | 84 |
| ManiSkill2: TurnFaucet w/g (State) | 2 | 64 |
| ManiSkill2: TurnFaucet w/o g (State) | 2 | 72 |
| ManiSkill2: PushChair w/g (State) | 2 | 56 |
| ManiSkill2: PushChair w/o g (State) | 2 | 48 |
| Adroit: Door (State) | 2 | 74 |
| Adroit: Pen (State) | 2 | 4 |
| Adroit: Hammer (State) | 2 | 32 |
| Adroit: Relocate (State) | 2 | 84 |

Table 12: The number of our RNR-DP trainable parameters for each task. Noise-relaying buffer size doesn't affect the number of trainable parameters for each task.

| Task | Trainable Params |
|---|---|
| ManiSkill3: PushT (Visual) | 11.42M |
| ManiSkill3: RollBall (Visual) | 11.59M |
| ManiSkill2: StackCube (Visual) | 10.85M |
| Adroit: Pen (Visual) | 14.67M |
| Adroit: Hammer (Visual) | 14.72M |
| Adroit: Door (Visual) | 14.41M |
| | |
| ManiSkill3: PushT (State) | 4.53M |
| ManiSkill3: RollBall (State) | 4.73M |
| ManiSkill2: StackCube (State) | 4.91M |
| ManiSkill2: TurnFaucet (State) | 4.71M |
| Adroit: Door (State) | 4.66M |
| Adroit: Pen (State) | 4.75M |
| Adroit: Hammer (State) | 4.77M |
| Adroit: Relocate (State) | 4.66M |

# C   Additional Results

## C.1   Empirical Comparison with Acceleration Methods on Visual Observations

We summarize the results of vision-based experiments in Table 15. As shown in Table 15, our RNR-DP ourperforms all DDIM variations and CP variations and particularly has an overall improvement over 8-step DDIM by 6.9%, over 8-step-chaining CP by 3.4%.

Table 13: We list the key hyperparameters of Diffusion Policy baseline used in our experiments.

| Hyperparameter | Value |
|---|---|
| Diffusion Step Embedding Dimension | 64 |
| UNet Downsampling Dimensions | [64, 128, 256] |
| Optimizer | AdamW |
| Weight Decay | 1e-6 |
| Learning Rate | 1e-4 |
| Learning Rate Scheduler | Cosine |
| EMA Model Update | 0.9999 |
| Online Evaluation Episodes | 1000 |

Table 14: We list the observation horizon, action execution horizon and action prediction horizon of Diffusion Policy baseline for each task.

| Task | Obs $T_o$ | Act Exec $T_a$ | Act Pred $T_p$ |
|---|---|---|---|
| ManiSkill3: PushT (Visual) | 2 | 2 | 16 |
| ManiSkill3: RollBall (Visual) | 2 | 4 | 16 |
| ManiSkill2: StackCube (Visual) | 2 | 8 | 16 |
| Adroit: Pen (Visual) | 2 | 8 | 16 |
| Adroit: Hammer (Visual) | 2 | 8 | 16 |
| Adroit: Door (Visual) | 2 | 8 | 16 |
| ManiSkill3: PushT (State) | 2 | 1 | 16 |
| ManiSkill3: RollBall (State) | 2 | 4 | 16 |
| ManiSkill2: StackCube (State) | 2 | 8 | 16 |
| ManiSkill2: TurnFaucet (State) | 2 | 8 | 16 |
| Adroit: Door (State) | 2 | 8 | 16 |
| Adroit: Pen (State) | 2 | 8 | 16 |
| Adroit: Hammer (State) | 2 | 8 | 16 |
| Adroit: Relocate (State) | 2 | 8 | 16 |

Table 15: Evaluation on simpler tasks (Regular Group) not requiring responsive control from ManiSkill and Adroit benchmarks (Visual Observations). We follow our evaluation metric and report values under the same settings as in Table 4. Tasks in which none of the methods achieve a reasonable success rate under visual observations are omitted.

| Method | Steps (S) | NFEs/a | StackCube | Pen | Hammer | Avg. SR of tasks |
|---|---|---|---|---|---|---|
| DDPM | 100 | 12.5 | 0.958 | 0.133 | 0.123 | 0.404 |
| DDIM | 1 | 0.125 | 0.000 | 0.000 | 0.000 | 0.000 |
| | 2 | 0.25 | 0.946 | 0.042 | 0.000 | 0.329 |
| | 4 | 0.5 | **0.947** | 0.125 | 0.000 | 0.357 |
| | 8 | 1 | 0.946 | 0.139 | 0.009 | 0.365 |
| EDM | 80 | 20 | 0.930 | 0.156 | 0.067 | 0.384 |
| CP | 1 | 0.125 | 0.615 | 0.127 | 0.088 | 0.277 |
| | 8 | 1 | 0.910 | **0.161** | 0.077 | 0.383 |
| RNR-DP | 1 | 1 | 0.924 | 0.154 | **0.110** | **0.396** |

### C.2 Comparsion with Streaming Diffusion Policy (SDP)

Streaming Diffusion Policy (SDP) (Høeg et al., 2024) is a recent advancement over Diffusion Policy that stays close to our approach. In this section, we compare our method with SDP in terms of motivation (Appendix C.2.1), methodology (Appendix C.2.2), and empirical results (Appendix C.2.3).

#### C.2.1 Motivation Comparison

SDP accelerates Diffusion Policy inference by reducing the number of denoising steps required to generate an action sequence. While improving diffusion inference speed is a relevant research topic, its impact in robotics is less compelling, as DDIM and Consistency Policy already provide reasonable speedups with strong performance. In contrast, our method addresses a fundamental limitation of Diffusion Policy—its lack of responsiveness—which significantly hinders performance in rapidly changing environments (e.g., contact-rich dynamic object manipulation). This challenge is far more critical to advancing robotic control. Although our approach also serves as an effective acceleration method, we view this as a secondary benefit compared to its primary advantage of enabling more responsive control.

#### C.2.2 Method Comparison

**Rollout Method** SDP also employs an action buffer structure but partitions the prediction horizon $T_p$ into multiple action chunks, ensuring that (1) each chunk maintains the same noise level and (2) noise levels increase across chunks. This chunk-wise design focuses solely on reducing denoising steps and accelerating inference. However, it does not address responsiveness and thus retains the limitations of Diffusion Policy. In contrast, our sequential denoising scheme conditions all actions on the latest observations, enabling responsive control while leveraging the noise-relaying buffer to maintain efficiency.

**Policy Architecture** SDP fuses all time embeddings along the temporal dimension into a single embedding. In contrast, our architecture retains multiple time embeddings, ensuring that noisy actions within the noise-relaying buffer can perceive time step changes based on the latest observation features. This design preserves temporal dynamics, allowing each action to adapt to varying time steps, thereby improving responsiveness and consistency in action generation.

#### C.2.3 Empirical Results Comparison

To comprehensively compare Diffusion Policy, Streaming Diffusion Policy, and our method, we conduct experiments on both response-sensitive tasks and regular tasks to evaluate their responsiveness and efficiency, as shown in Table 16 and Table 17. The empirical results indicate that on response-sensitive tasks, Streaming Diffusion Policy performs similarly to Diffusion Policy, whereas our method achieves significantly more responsive control than both. On regular tasks, both SDP and our method successfully preserve the performance of DP, but our method achieves 6.25 times faster inference.

Table 16: We compare Diffusion Policy, Streaming Diffusion Policy with our method on response-sensitive tasks.

| Task | DP | SDP | RNR-DP |
|---|---|---|---|
| Relocate (Adroit) | 0.422 | 0.436 | **0.585** |
| PushChair w/ g (ManiSkill2) | 0.495 | 0.500 | **0.547** |
| PushChair w/o g (ManiSkill2) | 0.635 | 0.633 | **0.694** |

## D Task Grouping Discussion

In this section, we provide a detailed discussion of the criteria used for grouping tasks into response-sensitive tasks and regular tasks. Appendix D.1 outlines the general rules for task grouping, while Appendix D.2 explains the specific reasoning behind the classification of each task.

Table 17: We compare Streaming Diffusion Policy with Diffusion Policy and our method on regular tasks.

| Method | Steps (S) | NFEs/a | StackCube | TurnFaucet (w/ g) | TurnFaucet (w/o g) | Hammer |
|---|---|---|---|---|---|---|
| DP | **100** | **12.5** | **0.960** | **0.495** | **0.595** | **0.120** |
| SDP | 50 | 6.25 | **0.961** | 0.480 | **0.615** | 0.136 |
| RNR-DP | 1 | 1 | 0.935 | **0.531** | 0.594 | **0.139** |

### D.1 General Rules for Task Grouping

For response-sensitive tasks, inaccurate actions based on outdated observations can easily lead to states not covered by the demonstration dataset. For example, if the robot pushes a swivel chair too forcefully, it may fall and become unrecoverable. In contrast, regular tasks are more tolerant of inaccuracies from outdated observations. For instance, in the stack cube task, even if the gripper doesn't precisely stop above the cube, this state still falls within the distribution.

### D.2 Detailed Task Separation Criteria for Each Task

**Relocate (Adroit) (Response-Sensitive)** This task requires controlling a high-dimensional dexterous hand to pick up a ball from a surface and transport it to a goal position. Due to the potential for the ball to slip on the surface or shift unpredictably within the fingers, responsive real-time feedback is essential for successful execution.

**Door (Adroit) (Response-Sensitive)** This task requires controlling a high-dimensional dexterous hand to apply the appropriate force to turn the handle. Insufficient force fails to open the door, while excessive force causes the hand to slip off. Since each phase relies on the precise execution of the previous one, real-time control is essential for success.

**PushChair (ManiSkill2) (Response-Sensitive)** This task requires controlling a mobile bimanual system to push a swivel chair to a goal position. Effective force control is crucial—pushing too hard can cause the chair to topple over, making recovery impossible. Precisely stopping the chair at the goal position demands highly adaptable control to make fine adjustments. Without real-time feedback, the chair tends to bounce around the goal position instead of coming to an exact stop.

**RollBall (ManiSkill3) (Response-Sensitive)** This task requires precisely rolling a ball to reach the goal position. Successful execution depends on applying the appropriate force and accurately controlling the rolling trajectory. Even a slight control error can result in missing the target, making highly adaptable and responsive control essential.

**PushT (ManiSkill3) (Response-Sensitive)** This task involves pushing a T-shaped block to a goal region. Successful completion requires the block to cover 90% of the goal area, necessitating real-time fine adjustments to its position. Without adaptable control, precisely aligning the T-shaped block to the goal position becomes challenging and prone to failure.

**StackCube (ManiSkill2) (Regular)** This task requires controlling a gripper to pick up a cube and stack it onto another cube. It is tolerant to minor inaccuracies in control, as the gripper does not need to stop precisely above the cube; slight deviations still fall within the expected distribution.

**TurnFaucet (ManiSkill2) (Regular)** This task requires controlling a gripper to turn on a faucet. Since it involves simple manipulation without complex contact-rich operations, and the faucet remains fixed in place, it is tolerant to inaccuracies in control and does not require real-time adjustments.

**Pen (Adroit) (Regular)** This task involves controlling a high-dimensional dexterous hand to rotate a pen and align it with a goal pose. Successful completion in the dataset requires an average of only 30 steps, with a minimum of 13 steps, indicating a very short-horizon control requirement. Additionally, numerous studies (Nakamoto et al., 2023; Florence et al., 2022) have identified it as one of the easiest tasks in Adroit.

**Hammer (Adroit) (Regular)** This task involves controlling a high-dimensional dexterous hand to strike a nail. The hammering motion does not require fine-grained, real-time adjustments, as the primary objective is to deliver sufficient force to the nail. Minor deviations in trajectory or impact position do not significantly impact the task's success.

## E    Complex Real-world Scenarios Discussion

RNR-DP's responsiveness is crucial in complex real-world scenarios, where dynamic environments, disturbances, and multi-object interactions create significant variability. The ability to quickly adapt to changing conditions is key to achieving robust performance. This responsiveness could enable RNR-DP to tackle real-world challenges like object variations, complex interactions, human disturbances, and environmental uncertainties, broadening its applicability to a wider range of tasks.

We believe RNR-DP offers significant advantages for real-robot deployment. First, it provides responsive control to handle rapid environmental changes, which are crucial in both simulation and real-world scenarios. Second, in real-robot settings, the control policy must meet a specific control rate, unlike in simulation where the environment can wait for policy inference. While DP requires acceleration via DDIM for real-robot deployment, as noted in the DP paper, this does not preserve performance well in our experiments. RNR-DP, on the other hand, better preserves the DP performance while achieving the necessary control rate for real-robot applications.

## F    Multi-Modality Property of RNR-DP

To demonstrate the preservation of multi-modality in practice, we visualize action distributions from a state in the ManiSkill2 StackCube task, using our RNR-DP. We sample 1000 actions from our policy then applies PCA dimensionality reduction for visualization. We use histograms to visualize the discrete relative density of these action samples and use kernel density estimation (KDE) to visualize the estimated probability density function. The results are shown in Figure 16.

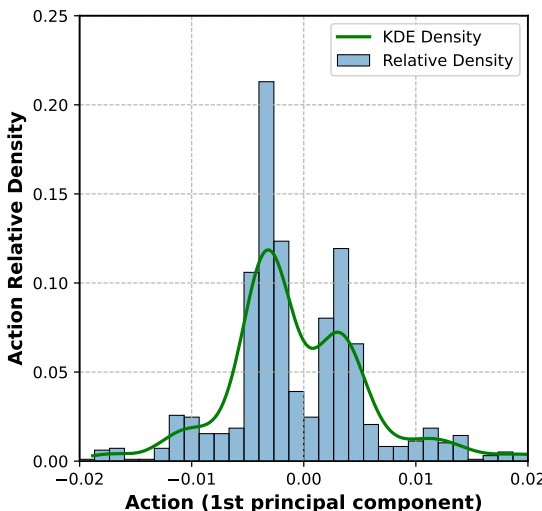

Figure 16: Conditional action distribution visualization from our experiments. Our method demonstrates clear bimodal distribution, showing that RNR-DP preserves multi-modality property.

We can see that the dimension-reduced action distribution exhibits a clear bimodal distribution, confirming that multi-modality is preserved in our method.

## G    Noise-Relaying Buffer Capacity Discussion

We find that the optimal buffer capacity is closely related to the task horizon, or the number of steps required to complete the task in the dataset. The rationale is that if a task is solved in 30 steps, an 84-step noise-relaying buffer would be inappropriate. Notably, Adroit Pen has the shortest task horizon, an average of 30 steps with a minimum of 13 steps, significantly lower than other tasks, and thus requires a much smaller buffer capacity. In practice, we set the buffer capacity to a value lower than the minimum task horizon in the dataset and make adjustments around this value. For every task, including Adroit Pen, the buffer capacity has a wide range of workable values.

## H    Noise Scheduling Discussion

In this section, we aim to discuss two noise scheduling method from Streaming Diffusion Policy (Høeg et al., 2024), 67% Independent 33% Linear and Chunk-Wise Noise Scheduling.

### H.1    67% Independent 33% Linear Noise Scheduling

The key difference between Mixture Noise Scheduling and the "67% Independent, 33% Linear" schedule is the proportion of linear noise and random (or independent) noise. In our setup, we use 60% random noise (or independent noise from the SDP paper) and 40% linear noise from TEDi (Zhang et al., 2024), which has proven effective and robust across all tasks. We also tested the "67% independent (random), 33% linear" noise schedule from SDP (Høeg et al., 2024) for the Adroit Relocate task, and the results show it performs well, as the two noise schedules (67-33 vs. 60-40) are quite similar. Refer to Table 18 for detailed results.

### H.2    Chunk-Wise Noise Scheduling

To leverage current observations, RNR-DP uses a single-action rollout during each inference ($T_a = 1$). Chunk-wise noise scheduling is inappropriate for our setting (Table 18), as it is designed for action sequence rollouts (e.g., $T_a = 8$ in DP and SDP). Our scheduling is essentially a special case of chunk-wise noise perturbation, where the chunk size is set to 1.

Table 18:    Ablation study on noise scheduling scheme during training. Numbers represent average success rates ($\uparrow$).

| Ablation | Relocate (Adroit) |
|---|---|
| Chunk-Wise | 0.001 |
| 100% Linear | 0.323 |
| 100% Random (Independent) | 0.389 |
| 67% Independent (Random) 33% Linear | 0.558 |
| Mixture (60% Random 40% Linear) (Ours) | **0.585** |

## I    Mixture Scheduling Visualization

In this section, we provide a detailed discussion of the mechanism and motivation behind *Mixture Noise Scheduling*. As illustrated in Figure 17, the random schedule is teaching the model to denoise actions independently, where each action is assigned a random noise level. The linear schedule, on the other hand, maintains an increasing noise level across actions, closely aligning with our inference process through the noise-relaying buffer. Our mixture schedule not only trains the model to denoise actions independently, as in the DP setting, but also ensures smooth transitions between consecutive actions. This better aligns with the noise-relaying buffer structure, resulting in more diverse and robust trainings.

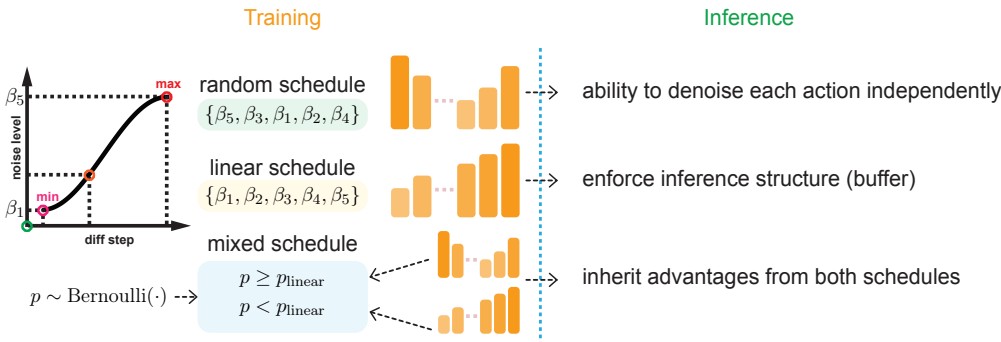

Figure 17: Detailed illustration of the process of Mixture Scheduling.

## J   Laddering Initialization Visualization

In this section, we provide a clear and concise discussion of *Laddering Initialization*. As illustrated in Figure 18, to transition from random noise to an increased noise level suitable for inference through the noise-relaying buffer, we perform several denoising steps. This process results in a buffer with *laddered noise*, ensuring a more structured and effective initialization.

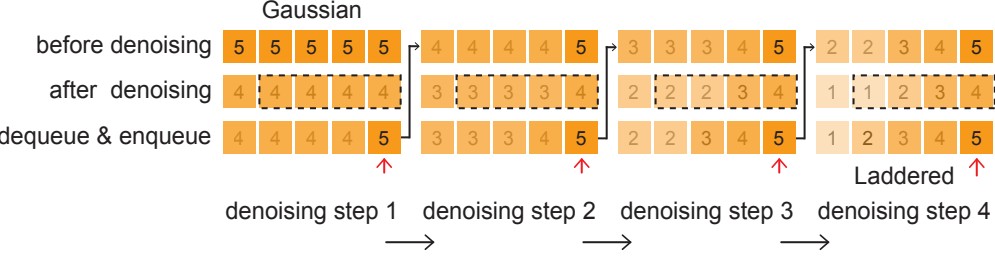

Figure 18: Detailed illustration of the process of Laddering Initialization. Everything happens before our policy interacts with the environment.

## K   Qualitative Examples of How DP Struggles with Response-Sensitive Tasks and How RNR-DP Resolves Them

Comparison videos of response-sensitive tasks between DP and RNR-DP can be found on this page. The videos clearly demonstrate that while DP struggles with responsive control, RNR-DP effectively completes the tasks with greater precision and adaptability.

## L   Discussion on Robosuite Tasks

We initially decided not to experiment with Robosuite for two reasons: (1) nearly all tasks can be solved by DP with close to 100% success, as demonstrated in the DP paper, and (2) the tasks primarily involve pick-and-place, which does not require highly responsive control.

## M   Response Sensitivity Analysis with RL Policies

In this section, we assess response-sensitivity through RL training and further validate our task classification. We design two experimental settings. In *Action Horizon* setting, RL agent directly outputs the next $T_a$ actions (as in Diffusion Policy); In *Action Repeat* setting, RL agent outputs one action which is then executed in the environment repeatedly for $T_a$ steps (a strong challenge to responsiveness). We experiment with

Adroit Relocate (response-sensitive), Adroit Pen (regular), and ManiSkill2 StackCube (regular) using SAC algorithm.

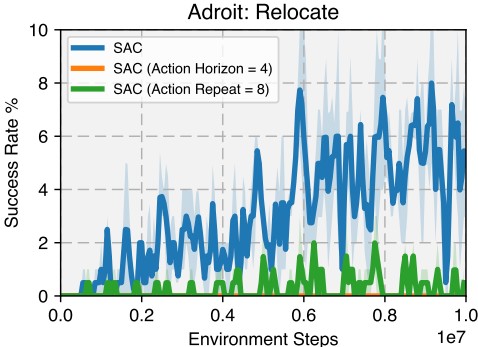

Figure 19: RL training results on Adroit Relocate (classified as response-sensitive task). Both the *Action Horizon* and *Action Repeat* experiments fail to achieve any success on Adroit Relocate, indicating that the task has very low tolerance to outdated actions and strongly depends on the most recent observations. This supports our classification of Adroit Relocate as a response-sensitive task.

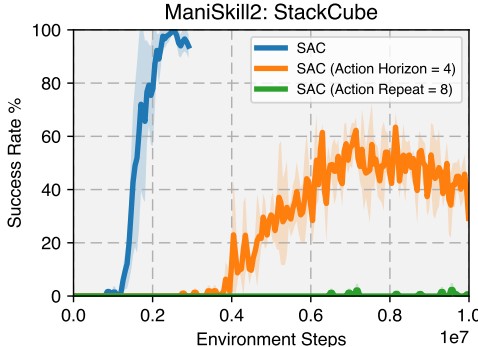

Figure 20: RL training results on ManiSkill2 StackCube (classified as regular task). *Action Horizon* runs can achieve decent success rate, while *Action Repeat* runs cannot, indicating that ths task has some tolerance to outdated actions. This supports our classification of ManiSkill2 StackCube as a regular task.

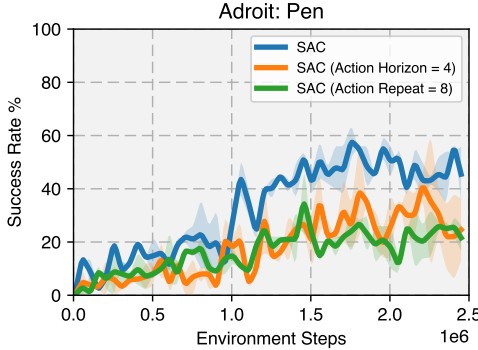

Figure 21: RL training results on Adroit Pen (classified as regular task). Both *Action Horizon* and *Action Repeat* runs have reasonable success rate, indicating that the task has very high tolerance to outdated actions. This supports our classification of Adroit Pen as a regular task.

In the *Action Horizon* setting, $T_a = 8$ is unsolvable due to the large action space, which hinders RL exploration. Behavior Cloning handles this better since it's supervised. Hence, we focus on $T_a = 4$, where Adroit Pen and ManiSkill2 StackCube achieve reasonable success, while Adroit Relocate struggles and achieves nearly-zero success rate (See Figures 19 to 21). In the *Action Repeat* setting, repeating the same action for $T_a = 8$ steps poses a high responsiveness challenge. Even so, Adroit Pen still achieves some success, showing strong tolerance to outdated actions (See Figure 21). In summary, these experiments confirm that Adroit Pen and ManiSkill2 StackCube tolerate outdated actions better than Adroit Relocate, supporting our task classification.

