# OpenReview forum: "Responsive Noise-Relaying Diffusion Policy: Responsive and Efficient Visuomotor Control"
_TMLR — Accepted by TMLR_

### Review · Reviewer_rwF3 · 2025-02-24

**Summary Of Contributions:**

This paper outlines the two problems when using diffusion policies for robotic control: (1) Limited policy responsiveness due to large action horizon, and (2) with small action horizon, diffusion policy exhibits mode bouncing where consecutive actions, computed at different model inference steps, may come from different modalities. This paper proposes a method, RNR-DP, that aims to address both problems by incorporating a noise-relaying buffer.
This noise-relaying buffer incorporates partially denoised actions for future timesteps, so that the policy can perform less denoising steps.
Furthermore, the partially denoised actions include information about the current action, thereby enabling the policy to ensure future actions come from the same mode.
Finally, this paper provides an empirical example to demonstrate the mode-bouncing phenomenon, and further conduct experiments to show the improved responsiveness and improved computation efficiency when compared to existing methods such as DDPM, DDIM, EDM, etc.

**Audience:**

Yes

**Broader Impact Concerns:**

As far as I know there isn't any broader impact concerns.

**Claims And Evidence:**

No

**Requested Changes:**

**Changes for Securing Recommendation**
- The term "responsiveness" can be broken down into whether the current action applied is leveraging the current observation, or the inference time. The paper should clearly define what responsiveness means here, otherwise it might be misleading to future readers.
- The writing on section 5.2 in particular appears to be critical in this paper, but I have trouble understanding ladder initialization in particular. If the paper can clarify how we started with standard Gaussian to "laddered noise" with random schedule that should address it.
- Experiments appear to have limited evaluations---for an empirical paper that only considers simulated environment, I believe the analyses need to be more rigorous.
- There should be an explanation on why specific task is in particular task group---based on the task description in the appendix I am not convinced that some tasks in the "non-sensitive group", e.g., pick-and-place is not any more sensitive, are less sensitive to ones in the "sensitive group". Some other environment recommendations such as dynamic scene manipulation will be more convincing.

**Strengths And Weaknesses:**

**Strengths**
- The diffusion policy responsiveness problem is well motivated and is of interest of robot learning practitioners---resolving the responsiveness problem of diffusion policies will enable wider range of robotic applications, especially manipulation in dynamic scenes.
- The noise-relaying buffer, if I understand correctly, provides a mechanism to ensure future actions come from the same mode since the backward process uses the partially denoised action from previous step.
- The ablation experiments seem to suggest that the noise-relaying buffer does improve the performance of the policy in the considered environments.

**Major Comments**

***Writing***
- I believe the writing can be further improved
	- Page 1, abstract, "... executing multiple actions to retain performance..., which limits its responsiveness, ...": It is unclear that multiple actions here are coming from the same inference step.
	- Page 1, abstract, "Further evaluation on regular tasks demonstrates that RNR-DP also exceeds the best acceleration method...": Here it is unclear what RNR-DP is exceeding.
	- Figure 1: What is the y-axis?
	- Page 3, background: $\beta$ is not shown in this paper and it seems to be $\beta = \gamma$ here?
	- More in the points below.

***Conceptual***
- There seems to be two types of acceleration: (1) on inference time, and (2) on action horizon. Some of the existing methods aims to speed (1) up but may not necessarily improve (2), and vice versa. Can this paper outline these for clarity?
	- Furthermore, even if the improvement is on (2), one of the current limitations of diffusion policies appear to be large sampling time, resulting in poor policy responsiveness---does the paper improve this? If so, I believe this will greatly improve the quality of the paper. Otherwise, I would love the paper to clarify this.
- Regarding section 5.1: While I believe that noise-relaying buffer can help alleviate mode bouncing, I believe it depends on whether the model is "autoregressive", in the sense that, e.g., conditioning on $a_{<t}$ for $a_t$, like diffusion policy did. Can the paper clarify if this is necessary?
- Regarding section 5.2: I have some trouble understanding the key design choices
	- Mixture Noise Scheduling: My understanding is that linear schedule is to essentially trying to enforce structure similar to inference (which is to alleviate mode bouncing), while random schedule is more similar to standard diffusion model training (which is to help noise-relaying buffer initialization).
	- Ladder Initialization: The description here is not really clear to me. Does that mean, for every $a_f$ the model runs the backward step $f$ times with random variance each step, rather than following the linear schedule? How does this align to Figure 3 where it says, "the buffer is denoised for $f$ times until the buffer head contains minimal noise level." How exactly does the buffer head contain minimal noise level and how is that determined?

***Experiments***
- Are all of these experiments executed with only a single seed each? If so, doesn't few percentage improvement in the success rate very low, and potentially statistically insignificant?
- Regarding task descriptions,I have trouble understanding why the task groups are separated this way---how are pick-and-place and adroit pen not response sensitive tasks? Are they not dynamic? Perhaps for tasks such as pushing and pick-and-place, the environments consider the force applied onto the object and enforce the force to be below some threshold---here I suspect it's more clear that the model needs to be response sensitive. Alternatively, one may include a task where the agent needs to place item onto a moving platform.

**Minor Comments**
- Section 4.2, on the claim that diffusion policy struggles with dynamic, contact-rich manipulation tasks: While I agree this is likely the case, is there anywhere on the manuscript that supports this? If so, could the paper refer to it?
- Based on the amount of visual-observation experiments done for this paper, I don't completely see that this work applying only to visuomotor control (which is a good thing in my opinion), but perhaps I am missing something here?

---

> ### Author Response · Authors · 2025-03-15
> **Response to Reviewer rwF3 (1/3)**
>
> We sincerely appreciate the reviewer’s constructive feedback and have addressed the comments in detail below.
>
> ***
>
> # Major Comments
>
> ## Writing
>
> **[Q1]** *Clarification of abstract, figure labeling and symbol notation*
>
> > …It is unclear that multiple actions here are coming from the same inference step
>
> We have revised the abstract to make it clearer that diffusion policy executes multiple actions predicted from the same inference time step to retain performance.
>
>
> > …it is unclear what RNR-DP is exceeding
>
> We have revised the `abstract` to make it clear that RNR-DP exceeds the best acceleration method (DDIM) in terms of success rate.
>
>
> > What is the y-axis in Figure 1?
>
> In Figure 1, both the y-axis of `Response-sensitive Tasks` and `Regular Tasks` are the success rate. We updated `Figure 1` to address this confusion.
>
>
> > background: β is not shown in this paper and it seems to be β=γ here?
>
> Thanks for pointing out this notation typo, and we have fixed it.
>
> ## Conceptual
>
> **[Q2]** *Clarification of two types of acceleration and classify RNR-DP’s acceleration type*
>
> > There seems to be two types of acceleration: (1) on inference time, and (2) on action horizon. Some of the existing methods aim to speed (1) up but may not necessarily improve (2), and vice versa. Can this paper outline these for clarity?
> >
> > …even if the improvement is on (2), one of the current limitations of diffusion policies appears to be large sampling time…does the paper improve this?…
>
> RNR-DP uses a single-action rollout to leverage current observations and provide responsive control, while also reducing diffusion sampling time. Although it does not use advanced diffusion solvers like Consistency Policy to directly shorten the sampling time, RNR-DP reduces the number of denoising steps needed to generate an action by employing a noise-relaying buffer. This buffer allows the system to reuse outputs from previous diffusion inferences, requiring only one denoising step to generate an action at each inference.
>
> **[Q3]** *Clarification of noise-relaying buffer’s dependency on autoregressive structure*
>
> > Regarding section 5.1: While I believe that noise-relaying buffer can help alleviate mode bouncing, I believe it depends on whether the model is "autoregressive", in the sense that, e.g., conditioning on a<t for at, like diffusion policy did. Can the paper clarify if this is necessary?
>
> The noise-relaying buffer mitigates mode bouncing by ensuring that each action is generated sequentially and remains strongly correlated with previous actions, unlike Diffusion Policy, where entire action sequences are sampled independently from random noise. If this characteristic aligns with the reviewer’s definition of *autoregressive*, then our approach would indeed need to be considered *autoregressive*.
>
> **[Q4]** *Confusion of two schedules used in Mixture Noise Scheduling*
>
> > Mixture Noise Scheduling: My understanding is that linear schedule is essentially trying to enforce structure similar to inference (which is to alleviate mode bouncing), while random schedule is more similar to standard diffusion model training (which is to help noise-relaying buffer initialization).
>
> We believe you’ve captured our point! The random schedule is teaching the model to denoise actions independently, where each action is assigned a randomized noise level. The linear schedule, on the other hand, maintains an increasing noise level across actions, closely aligning with our inference process through the noise-relaying buffer. Our mixture schedule not only trains the model to denoise actions independently, but also ensures smooth transitions between consecutive actions. This better aligns with the noise-relay buffer structure, resulting in more diverse and robust training. Refer to `Appendix I` for illustration figure and detailed discussion. The results are also shown in [this link](https://sites.google.com/view/rnr-dp/home/more-visualizations?authuser=0#h.zfiz6gtbp9xi).

---

> > ### Author Response · Authors · 2025-03-15
> > **Response to Reviewer rwF3 (2/3)**
> >
> > **[Q5, Change2]** *Confusion of the technical detail of Laddering Initialization*
> >
> > > Ladder Initialization: The description here is not really clear to me. Does that mean, for every $a_f$ the model runs the backward step $f$ times with random variance each step, rather than following the linear schedule? How does this align to Figure 3 where it says, "the buffer is denoised for $f$ times until the buffer head contains minimal noise level." How exactly does the buffer head contain minimal noise level and how is that determined?
> >
> > > ...clarify how we started with standard Gaussian to "laddered noise" with random schedule...
> >
> > To transition from random noise to an increased noise level suitable for inference through the noise-relaying buffer, we perform several denoising steps. This process results in a buffer with "laddered noise," ensuring a more structured and effective initialization. Refer to Appendix J for illustration figure and detailed discussion. The results are also shown in [this link](https://sites.google.com/view/rnr-dp/home/more-visualizations?authuser=0#h.hnz37mynl21d).
> >
> > **[Change 1]** *Define responsiveness clearly*
> > > The term "responsiveness" can be broken down into whether the current action applied is leveraging the current observation, or the inference time. The paper should clearly define what responsiveness means here, otherwise it might be misleading to future readers.
> >
> >
> > We have revised the manuscript to clearly define "responsiveness" in `introduction` based on your feedback. In our paper, we define responsiveness as the degree to which the current action leverages the most recent observation. This definition highlights the model's ability to adapt its decisions based on the latest information, rather than relying on outdated or precomputed responses.
> >
> > ## Experiment
> >
> > **[Q6, change3]** *Number of seeds used for experiments*
> > > Are all of these experiments executed with only a single seed each? If so, doesn't few percentage improvement in the success rate very low, and potentially statistically insignificant?
> >
> > > Experiments appear to have limited evaluations---for an empirical paper that only considers simulated environments, I believe the analyses need to be more rigorous.
> >
> > As stated in the captions of `Table 2`, all experiments were conducted with 10 random seeds, evaluating 100 episodes per seed, totaling 1000 episodes. We report the success rate based on this evaluation. Our experimental setup is more rigorous than many well-known imitation learning papers, including BeT[1] and IBC[2].
> >
> > [1] Shafiullah, Nur Muhammad, et al. "Behavior transformers: Cloning $ k $ modes with one stone." Advances in neural information processing systems 35 (2022): 22955-22968
> >
> > [2] Florence, Pete, et al. "Implicit behavioral cloning." Conference on robot learning. PMLR, 2022
> >
> > **[Q7, change4]** *Justification of task group division*
> > > Regarding task descriptions,I have trouble understanding why the task groups are separated this way---how are pick-and-place and adroit pen not response sensitive tasks? Are they not dynamic? Perhaps for tasks such as pushing and pick-and-place, the environments consider the force applied onto the object and enforce the force to be below some threshold---here I suspect it's more clear that the model needs to be response sensitive. Alternatively, one may include a task where the agent needs to place an item onto a moving platform.
> >
> > > There should be an explanation on why specific task is in particular task group---based on the task description in the appendix I am not convinced that some tasks in the "non-sensitive group", e.g., pick-and-place is not any more sensitive, are less sensitive to ones in the "sensitive group". Some other environment recommendations such as dynamic scene manipulation will be more convincing.
> >
> > For `response-sensitive tasks`, inaccurate actions based on outdated observations can easily lead to states not covered by the demonstration dataset. For instance, if a robot pushes a swivel chair too forcefully, it may topple over and become irrecoverable. In contrast, `regular tasks` are more forgiving of inaccuracies from outdated observations. We classify Stack Cube (our only pick-and-place task) as a regular task because, even if the gripper does not stop precisely above the cube, the resulting state remains within the dataset distribution. We categorize Adroit Pen as a regular task due to its short horizon—averaging 30 steps with a minimum of 13 steps in the dataset—and its recognition in multiple studies as one of the easiest tasks in Adroit [1,2]. A detailed discussion of task groupings and classification criteria can be found in `Appendix D`.
> >
> > [1] Nakamoto, Mitsuhiko, et al. "Cal-ql: Calibrated offline rl pre-training for efficient online fine-tuning." Advances in Neural Information Processing Systems 36 (2023): 62244-62269
> >
> > [2] Florence, Pete, et al. "Implicit behavioral cloning." Conference on robot learning. PMLR, 2022

---

> > > ### Author Response · Authors · 2025-03-15
> > > **Response to Reviewer rwF3 (3/3)**
> > >
> > > # Minor Comments
> > >
> > >
> > > **[Q8]** *missing references of diffusion policy performance on dynamic tasks*
> > > > Section 4.2, on the claim that diffusion policy struggles with dynamic, contact-rich manipulation tasks: While I agree this is likely the case, is there anywhere on the manuscript that supports this? If so, could the paper refer to it?
> > >
> > > Comparison videos of response-sensitive tasks between DP and RNR-DP can be found on [this page](https://sites.google.com/view/rnr-dp/home/improvement-videos).
> > > The videos clearly demonstrate that while DP struggles with responsive control, RNR-DP effectively completes the tasks with greater precision and adaptability.
> > >
> > > **[Q9]** *Applicability scope beyond visuomotor control*
> > > > Based on the amount of visual-observation experiments done for this paper, I don't completely see that this work applies only to visuomotor control
> > >
> > > As our experiments include both state-based and visual-observation experiments, our method is capable of performing both state-based and visuomotor control. We titled our paper "Visuomotor Control" because visual-observation control is more crucial and has broader applicability in real-world scenarios.
> > >
> > > ***
> > >
> > > Hope the above clarifications and responses address your concerns. Thank you again for your time and effort!

---

> > > > ### Comment · Reviewer_rwF3 · 2025-03-21
> > > >
> > > > Thank you for the detailed response and clarifying my questions, particularly the term "responsiveness", the methodology with figures, and the experiment setting.
> > > >
> > > > I have also looked into other reviews and responses, and have few additional questions/comments:
> > > > 1. While I agree that having real-robot experiments will be nice, I think the paper can provide results on the inference speed in terms of wall time---since the method has less NFE/a, one would surmise that seconds/a is shorter too---can the paper verify that this is the case? If they don't positively correlate, roboticists will definitely find this an important detail.
> > > > 2. Regarding experiments, thank you for clarifying the number of trials and explaining why, e.g. Robosuite, is not considered. I believe it will be helpful to specify the standard error (e.g. table 2)---or perhaps explain what statistically better mean in this case.
> > > > 3. Regarding response-sensitive tasks, thank you for providing the details. Personally I still find it difficult to differentiate what they should correspond to---something more clear cut could be whether the agent can recover from outside the demonstration distribution, e.g. once the ball leaves the hand by accident or the chair falls then the agent can no longer recover. Alternatively one could measure whether one environment needs more "responsiveness than not" with additional experiments with RL-based agents that learn policies spanning $T_a$ steps.

---

> > > > > ### Author Response · Authors · 2025-03-28
> > > > > **Further Response to Reviewer rwF3**
> > > > >
> > > > > Thank you for your additional feedback. We address your comments in detail below. Since we are unsure whether editing the manuscript is currently permitted, we commit to including this discussion in the camera-ready version if the paper is accepted.
> > > > >
> > > > > ----
> > > > >
> > > > > **[Q1]** *Inference speed improvement in terms of wall time*
> > > > > > …I think the paper can provide results on the inference speed in terms of wall time---since the method has less NFE/a, one would surmise that seconds/a is shorter too---can the paper verify that this is the case?...
> > > > >
> > > > > We benchmark the inference time of Diffusion Policy and Responsive Noise-Relaying Diffusion Policy in terms of seconds per action (i.e., the seconds required to generate a single action). Each method is evaluated over 100 independent runs, and the average inference time is reported below.
> > > > >
> > > > >
> > > > > |                                  |   DP    | RNRDP  | NFEs/a (times) | Speed Improvement (times) |
> > > > > | -------------------------------- | :-----: | :----: | :---: | :-----------------------: |
> > > > > | Inference time needed per action | 35.76ms | 2.26ms | 12.5  |           15.82           |
> > > > >
> > > > >
> > > > >
> > > > > The results show that the inference time improvement between Diffusion Policy and Responsive Noise-Relaying Diffusion Policy is closely aligned with the `NFEs/a` improvement. This outcome is expected, as both methods use very similar U-Net architectures, making NFEs/a a reliable proxy for actual inference time (unit: `ms/a`).
> > > > >
> > > > >
> > > > > **[Q2]** *Statistical Significance of experiments*
> > > > > > I believe it will be helpful to specify the standard error…or perhaps explain what statistically better means in this case.
> > > > >
> > > > > We conduct our experiments using 10 random seeds to account for variability and avoid reliance on a "lucky" seed. Additionally, we run a total of 1000 episodes to capture a wide range of environment configurations, rather than benefiting from an "easily-solved" setup. These choices are intended to provide a more accurate and reliable estimation of each method’s performance. We report the standard deviation in success rates for selected tasks below.
> > > > >
> > > > >
> > > > > |                  |      DP       |       RNR-DP       |
> > > > > | ---------------- | :-----------: | :----------------: |
> > > > > | Relocate         | 0.422 ± 0.049 | **0.585** ± 0.041 |
> > > > > | Door             | 0.558 ± 0.043 | **0.629** ± 0.068  |
> > > > > | PushChair (w/ g) | 0.495 ± 0.042 | **0.547** ± 0.027  |
> > > > >
> > > > >
> > > > > Based on the results above, we clearly observe that  `mean_sr(RNR-DP) - std(RNR-DP)` > `mean_sr(DP)` and  `mean_sr(DP) + std(DP) < mean_sr(RNR-DP)`. This indicates that our experimental results and conclusions are statistically significant.
> > > > >
> > > > >
> > > > > **[Q3]** *More clear cut on task separation*
> > > > > > …something more clear cut could be whether the agent can recover from outside the demonstration distribution…additional experiments with RL-based agents that learn policies spanning Ta steps.
> > > > >
> > > > > Thank you for your suggestion! We believe that in `response-sensitive tasks`, inaccurate actions based on outdated observations can easily lead to states not covered by the demonstration dataset, from which the agent cannot recover. In contrast, for `regular tasks`, inaccuracies caused by outdated observations either do not result in OOD states or lead to OOD states that are easy for the agent to recover from.
> > > > >
> > > > > Following your suggestion to assess responsiveness through RL training, we design two experimental settings. The first, `action horizon`, has the agent output the next $T_a$ actions directly (as in Diffusion Policy), with each action being different. The second, `action repeat`, is a more extreme test—one action is output and repeated for $T_a$ steps. We run both experiments using the **SAC** algorithm on **Adroit Relocate (response-sensitive)**, **Adroit Pen (regular)**, and **ManiSkill2 StackCube (regular)**, focusing on the two regular tasks that raised the most concern. We use dense rewards, as sparse-reward RL is largely unsolvable. Full results are available [on this page](https://sites.google.com/view/rnr-dp/home/rl-policies?#h.yo7t96ayvpav).
> > > > >
> > > > > In the `action horizon` setting, $T_a = 8$ is unsolvable due to the large action space, which hinders RL exploration. Behavior Cloning handles this better since it's supervised. Hence, we focus on $T_a = 4$, where **Adroit Pen** and **ManiSkill2 StackCube** achieve reasonable success, while **Adroit Relocate** achieves none. In the `action repeat` setting, repeating the same action for $T_a = 8$ steps poses a high responsiveness challenge. Even so, **Adroit Pen** still achieves some success, showing strong tolerance to outdated actions.
> > > > >
> > > > > **In summary**, these experiments confirm that **Adroit Pen** and **ManiSkill2 StackCube** tolerate outdated actions better than **Adroit Relocate**, supporting our task classification.
> > > > >
> > > > > ----
> > > > > Hope the above clarifications and responses address your concerns. Thank you again for your time and effort!

---

> ### Comment · Reviewer_rwF3 · 2025-03-28
>
> Thank you for the updated results once again! Regarding the last point on action horizon/task responsiveness, I would argue StackCube can be seen as response-sensitive task as the degradation is pretty significant (~40% drop from $T_a = 1$ to $T_a = 4$). However, this threshold is likely different based on preferences---I believe this is more of a spectrum than a binary threshold. As long as the camera-ready paper reflects the extra sensitivity result and include a small discussion for it I am pretty convinced by the overall paper.

---

### Review · Reviewer_41NZ · 2025-03-03

**Summary Of Contributions:**

This paper presents a novel denoising diffusion process for robotic action prediction. They improve over the existing state-of-the-art, Diffusion Policy (DP), which predicts actions as “chunks” and rolls out parts of these chunks in an open-loop fashion. Such open-loop rollout makes DP less reactive to dynamic changes in the environment. This paper presents a solution that can help alleviate that issue by using a noisy action buffer where there is always a full-denoised action available at the head of the buffer to immediately utilize.

**Audience:**

Yes

**Broader Impact Concerns:**

No concerns.

**Claims And Evidence:**

Yes

**Requested Changes:**

1. Qualitative or quantitative experiments showing that the change proposed in this paper did not reduce the multi-modality in the action prediction. (critical)
2. Additional experiments on the same set of simulation environments as done in the DP paper. (strengthen)
3. Qualitative examples that could show that RNR-DP helps the model perform better at the response-sensitive group of tasks specifically due to higher reactivity. (critical)
4. Discussion around how this method could be beneficial in real-robot deployment. (critical)

**Strengths And Weaknesses:**

**Strengths:**
1. I find the paper well-written and easy to follow. The authors made it very clear in the introduction what the scope and key contributions of this paper are.
2. The motivation for proposing this method is made very clear in Section 4.1 and results in Table 1, which show that a longer action horizon in DP enables better performance in multi-modal tasks, and hence building a more responsive variant of DP that maintains the “action chunking” aspect is a meaningful thing to do.
3. I find results on the response-sensitive group of tasks (Tables 2 and 3) statistically significant and proof that this method does help improve policy reactiveness.

**Weaknesses:**
1. This paper aims to remove a limitation in Diffusion Policy, that is of reliance on a long action horizon for multi-modal action prediction, which reduces the reactiveness of the policy. However, I do not see any qualitative or quantitative experiments showing that the change proposed in this paper did not reduce the multi-modality in the action prediction. Hence, this is to ask, is RNR-DP strictly better than DP in terms of showing both multi-modal behavior and responsiveness, or, does it trade off multi-modality for responsiveness?
2. Since this paper ties closely with Diffusion Policy, additional experiments on the same set of simulation environments as done in the DP paper (from Robosuite) would greatly help a reader put RNR-DP in the same context.
3. There is a lack of qualitative examples that could show that RNR-DP helps the model perform better at the response-sensitive group of tasks specifically due to higher reactivity.
4. There is no discussion around how this method could be beneficial in real-robot deployment, if at all. There are also no real-robot experiments in this paper.

---

> ### Author Response · Authors · 2025-03-15
> **Response to Reviewer 41NZ**
>
> We sincerely appreciate the reviewer’s constructive feedback and have addressed the comments in detail below.
>
> ***
>
> **[Q1, Change1]** *Does RNR-DP trade off multi-modality for responsiveness*
>
> > …is RNR-DP strictly better than DP in terms of showing both multi-modal behavior and responsiveness, or, does it trade off multi-modality for responsiveness?
>
> > Qualitative or quantitative experiments showing that the change proposed in this paper did not reduce the multi-modality in the action prediction. (critical)
>
> To demonstrate that RNR-DP preserves multi-modality, we visualize action distributions from a state in the ManiSkill2 StackCube task using our RNR-DP. We sample 1000 actions from our policy and apply PCA for dimensionality reduction to aid visualization. We use histograms to show the discrete relative density of these action samples and kernel density estimation (KDE) to visualize the estimated probability density function. The results, shown in [Figure 16](https://sites.google.com/view/rnr-dp/home/more-visualizations#h.9weml3fmy3ps), clearly reveal a multi-modal distribution, demonstrating RNR-DP's preservation of multi-modality.
>
> These results are included in `Appendix F` of the updated manuscript.
>
> **[Q2, Change2]** *Additional Experiments on Robosuite*
> > additional experiments on the same set of simulation environments as done in the DP paper (from Robosuite).
>
> We initially decided not to experiment with Robosuite for two reasons: (1) nearly all tasks can be solved by DP with close to 100% success, as demonstrated in the DP paper, and (2) the tasks primarily involve pick-and-place, which does not require highly responsive control.
>
> **[Q3, Change3]** *Lack of qualitative examples of improvements at response-sensitive group*
>
> > There is a lack of qualitative examples that could show that RNR-DP helps the model perform better at the response-sensitive group of tasks specifically due to higher reactivity.
>
> Comparison videos of response-sensitive tasks between DP and RNR-DP can be found on [this page](https://sites.google.com/view/rnr-dp/home/improvement-videos).
> The videos clearly demonstrate that while DP struggles with responsive control, RNR-DP effectively completes the tasks with greater precision and adaptability.
>
>
> **[Q4, Change4]** *Lack of discussion of real-robot deployment*
>
> > There is no discussion around how this method could be beneficial in real-robot deployment, if at all.
>
> We believe RNR-DP offers significant advantages for real-robot deployment. First, it provides responsive control to handle rapid environmental changes, which are crucial in both simulation and real-world scenarios. Second, in real-robot settings, the control policy must meet a specific control rate, unlike in simulation where the environment can wait for policy inference. While DP requires acceleration via DDIM for real-robot deployment, as noted in the DP paper, this does not preserve performance well in our experiments. RNR-DP, on the other hand, better preserves DP performance while achieving the necessary control rate for real-robot applications. Refer to `Appendix E` for detailed discussion.
>
> ***
>
> Hope the above clarifications and responses address your concerns. Thank you again for your time and effort!

---

### Review · Reviewer_BUSv · 2025-03-05

**Summary Of Contributions:**

This paper identifies mode bouncing as a critical limitation in Diffusion Policy that compromises responsiveness and environmental adaptability. The authors propose RNR-DP (Noise-Relaying Diffusion Policy), which maintains temporal consistency through a noise-relaying buffer with progressively intensified noise levels while accelerating inference via sequential denoising conditioned on real-time observations. Extensive experimental validation confirms the method's simultaneous achievement of highly responsive control and operational efficiency.

**Audience:**

Yes

**Broader Impact Concerns:**

None.

**Claims And Evidence:**

Yes

**Requested Changes:**

1) To effectively showcase fast, responsive closed-loop control, it's essential to conduct real-robot experiments that include human disturbances and retry behaviors.
2) While Section C.2 provides some comparison to streaming diffusion policies, expanded comparisons across diverse tasks and simulations would strengthen the analysis.

**Strengths And Weaknesses:**

## Strength
1. The paper clearly defines the limitations of Diffusion Policy, establishing a strong motivation for the proposed RNR-DP approach. The analysis of how large action horizon requirements constrain responsiveness is well-structured and logically presented.

2. The method effectively decouples the denoising process along two dimensions: the action horizon (a₀–a_T) and the progressive denoising of individual actions (a_k). By reusing previously denoised actions, the approach achieves a 6.9% computational acceleration.

3. The final denoising step for each action incorporates current observation conditioning, enabling RNR-DP to achieve high responsiveness.

4. The experimental design encompasses diverse robotic tasks and observation modalities. Comparisons with established baselines (Diffusion Policy variants, Consistency Policy, etc.) demonstrate rigorous evaluation, while ablation studies systematically quantify component contributions.

## Weakness
1. The discussion of RNR-DP's applicability could be enhanced by extending its theoretical analysis to complex real-world scenarios involving multi-object interactions, dynamic environmental changes, and unmodeled disturbances.

2. The proposed Mixture Noise Scheduling mechanism appears to partition actions at p_linear=0.4 (Table 10), suggesting approximately 60% of actions receive random perturbations (p>0.4) while 40% undergo linear variance perturbations. This resemblance to the "67% Independent 33% Linear" configuration in SDP (Table 2) shows different performance characteristics compared to SDP's optimal "80% CW 20% Constant" setting. Could the authors clarify the difference between "Mixture Noise Scheduling" and "67% Independent 33% Linear" setting?

3. While Figure 5 demonstrates the performance sensitivity to noise-relaying buffer capacity, with authors indicating optimal ranges between 56-92 units, Appendix Table 11 reveals substantial cross-task variability in empirical observations (e.g., Adroit: Pen requiring 4 units vs. other tasks needing 84). Could the authors clarify the rationale for task-specific buffer capacity variations and the underlying factors contributing to its substantial impact?

4. As shown in Table 5, the authors ablate both linear and random perturbations. However, since SDP is closely related to this work, it would be helpful for the authors to ablate the chunk-wise noise perturbation.

---

> ### Author Response · Authors · 2025-03-15
> **Response to Reviewer BUSv (1/2)**
>
> We sincerely appreciate the reviewer’s constructive feedback and have addressed the comments in detail below.
>
> **[Q1]** *Discussion of complex real-world scenarios*
> > The discussion of RNR-DP's applicability could be enhanced by extending its theoretical analysis to complex real-world scenarios involving multi-object interactions, dynamic environmental changes, and unmodeled disturbances.
>
> RNR-DP's responsiveness is crucial in complex real-world scenarios, where dynamic environments, disturbances, and multi-object interactions create significant variability. The ability to quickly adapt to changing conditions is key to achieving robust performance. This responsiveness could enable RNR-DP to tackle real-world challenges like object variations, complex interactions, human disturbances, and environmental uncertainties, broadening its applicability to a wider range of tasks. We have added this discussion to `Appendix E`.
>
> **[Q2]** *Difference between Mixture Noise Scheduling and 67% Independent 33% Linear*
>
> > …Could the authors clarify the difference between "Mixture Noise Scheduling" and "67% Independent 33% Linear" setting?
>
> The key difference between Mixture Noise Scheduling and the "67% Independent, 33% Linear" schedule is the proportion of linear noise and random (or independent) noise. In our setup, we use 60% random noise (or independent noise from the SDP paper) and 40% linear noise from TEDI paper, which has proven effective and robust across all tasks. We also tested the "67% independent (random), 33% linear" noise schedule from the SDP paper for the Adroit Relocate task, and the results show it performs well, as the two noise schedules (67-33 vs. 60-40) are quite similar. Refer to our detailed discussion in `Appendix H`.
>
> **[Q3]** *Buffer capacity variation across tasks*
>
> > …Could the authors clarify the rationale for task-specific buffer capacity variations and the underlying factors contributing to its substantial impact?
>
> We find that the optimal buffer capacity is closely related to the task horizon, or the number of steps required to complete the task in the dataset. The rationale is that if a task is solved in 30 steps, an 84-step noise-relaying buffer would be inappropriate. Notably, Adroit Pen has the shortest task horizon, an average of 30 steps with a minimum of 13 steps, significantly lower than other tasks, and thus requires a much smaller buffer capacity. In practice, we set the buffer capacity to a value lower than the minimum task horizon in the dataset and make adjustments around this value. For every task, including Adroit Pen, the buffer capacity has a wide range of workable values. Refer to `Appendix G` for detailed discussion on noise-relaying buffer capacity.
>
> **[Q4]** *Ablation on chunk-wise noise perturbation*
>
> > As shown in Table 5, the authors ablate both linear and random perturbations. However, since SDP is closely related to this work, it would be helpful for the authors to ablate the chunk-wise noise perturbation.
>
> To leverage current observations, RNR-DP employs a single-action rollout during each inference (Ta = 1). Chunk-wise noise scheduling is unsuitable for our setting, as it is designed for action sequence rollouts (e.g., Ta = 8 in the DP and SDP papers). Our scheduling can be seen as a special case of chunk-wise noise perturbation, where the chunk size is set to 1. Despite its incompatibility, we conducted comparative experiments, and the results show that chunk-wise noise scheduling with the SDP setting (chunk_size = 8) achieves only 0.01% success on the Adroit Relocate task, proving that it does not align with RNR-DP. For a detailed discussion, please refer to `Appendix H`.

---

> > ### Author Response · Authors · 2025-03-15
> > **Response to Reviewer BUSv (2/2)**
> >
> > **[Change1]** *Real-robot experiments with human disturbances and retry behaviors*
> >
> > To enhance research reproducibility and promote broader accessibility, we conduct our experiments using widely adopted robotic simulation benchmarks. However, we believe RNR-DP offers significant advantages for real-robot deployment. First, it provides responsive control to handle rapid environmental changes, which are crucial in both simulation and real-world scenarios. Second, in real-robot settings, the control policy must meet a specific control rate, unlike in simulation where the environment can wait for policy inference. While DP requires acceleration via DDIM for real-robot deployment, as noted in the DP paper, this does not preserve performance well in our experiments. RNR-DP, on the other hand, better preserves DP performance while achieving the necessary control rate for real-robot applications. Refer to our detailed discussion on real-robot deployment in `Appendix E`.
> >
> > **[Change2]** *more comparison to Streaming Diffusion Policy*
> >
> > We extend the comparison between Diffusion Policy, Streaming Diffusion Policy, and our method to a broader range of response-sensitive and regular tasks to evaluate responsiveness and efficiency. The empirical results indicate that on response-sensitive tasks, Streaming Diffusion Policy performs similarly to Diffusion Policy, whereas our method achieves significantly more responsive control than both. On regular tasks, both SDP and our method successfully maintain the performance of DP, but our method achieves 6.25 times faster inference. This result is expected, as SDP primarily serves as an acceleration method rather than one aimed at enhancing responsiveness. Refer to `Appendix C.2` for complete results.
> >
> > ***
> > Hope the above clarifications and responses address your concerns. Thank you again for your time and effort!

---

### Comment · Action_Editor_xhD1 · 2025-03-13
**Author Discussion Phase**

Thank you very much to the reviewers for the detailed reviews.
We are now in the author rebuttal stage that runs unitil ~19th March.

Authors, please address any questions raised by the reviews.

Best,

AE

---

### Author Response · Authors · 2025-05-15
**Camera-Ready Revision Submitted**

Dear Action Editors and Reviewers,

---

We are pleased to inform you that we have completed and submitted the final camera-ready version of our paper.
We deeply appreciate your insightful comments, constructive feedback, and the time you dedicated to improving our work.

---

Best regards,

Authors

---

### Decision · Action_Editor_xhD1 · 2025-04-16

**Recommendation:** Accept as is

**Comment:**

All three reviewers recommend acceptance, acknowledging that the paper addresses an important problem in diffusion-based imitation learning policies. Initial concerns about task classification, hyperparameter sensitivity, missing comparisons, and real-world evaluation were adequately addressed in the author response. The authors provided additional visualizations, statistical analysis, and RL experiments to validate their approach.

While the paper lacks real-world robot experiments, the reviewers agreed this wasn't essential for acceptance given the comprehensive simulation results. The noise-relaying buffer mechanism is well-motivated and empirically validated across diverse robotic tasks. The authors have also demonstrated that their approach preserves the multi-modality capabilities of diffusion policies while improving responsiveness.

**Audience:**

TMLR's audience would be interested in this paper.
The work addresses a significant limitation in imitation learning approaches, particularly for robot control using diffusion models, which has gained substantial attention in recent years. The improvements in responsiveness and computational efficiency would be of interest to researchers working on diffusion models for control and robot learning.

**Claims And Evidence:**

The claims are supported by convincing evidence.
The authors demonstrate through extensive experiments that RNR-DP achieves an 18% improvement in success rate on response-sensitive tasks compared to Diffusion Policy, and exceeds the best acceleration method (DDIM) by 6.9% in success rate on regular tasks. Ablation studies effectively quantify component contributions, and all experiments were conducted with 10 random seeds, providing statistical significance.